# MaskPlace: Fast Chip Placement via Reinforced Visual Representation Learning

**Yao Lai**    **Yao Mu**    **Ping Luo** [*]
Department of Computer Science
The University of Hong Kong
{ylai,ymu,pluo}@cs.hku.hk

## Abstract

Placement is an essential task in modern chip design, aiming at placing millions of circuit modules on a 2D chip canvas. Unlike the human-centric solution, which requires months of intense effort by hardware engineers to produce a layout to minimize delay and energy consumption, deep reinforcement learning has become an emerging autonomous tool. However, the learning-centric method is still in its early stage, impeded by a massive design space of size ten to the order of a few thousand. This work presents MaskPlace to automatically generate a valid chip layout design within a few hours, whose performance can be superior or comparable to recent advanced approaches. It has several appealing benefits that prior arts do not have. Firstly, MaskPlace recasts placement as a problem of learning pixel-level visual representation to comprehensively describe millions of modules on a chip, enabling placement in a high-resolution canvas and a large action space. It outperforms recent methods that represent a chip as a hypergraph. Secondly, it enables training the policy network by an intuitive reward function with dense reward, rather than a complicated reward function with sparse reward from previous methods. Thirdly, extensive experiments on many public benchmarks show that MaskPlace outperforms existing RL approaches in all key performance metrics, including wirelength, congestion, and density. For example, it achieves 60%-90% wirelength reduction and guarantees zero overlaps. We believe MaskPlace can improve AI-assisted chip layout design. The deliverables are released at laiyao1.github.io/maskplace.

## 1 Introduction

The scalability and efficiency are two significant factors of autonomous chip layout design. Placement is one of the most challenging and time-consuming problems in the design flow, aiming to determine the locations of millions of circuit modules on a 2D chip canvas represented by a two-dimensional grid. A netlist can describe these modules, that is, a large-scale hypergraph consisting of massive macros (functional blocks such as memory) and standard cells (logic gates), where each macro and each standard cell can contain several or even hundreds of pins connected by wires, as shown in Fig.1.

Placing a large number of circuit modules onto the chip canvas is challenging because many performance metrics such as power consumption, timing, area, and wirelength should be minimized while satisfying some hard constraints such as placement density and routing congestion. For example, the wirelength (the length of wires that connect all modules) determines the delay and the power consumption of a chip [1]. Shorter wires often indicate less delay and less power consumption [2]. However, wirelength cannot be reduced by overlapping modules because the module density is a hard constraint to ensure that a valid and manufacturable chip layout has non-overlapping modules. More

---

[*]Corresponding author is Ping Luo

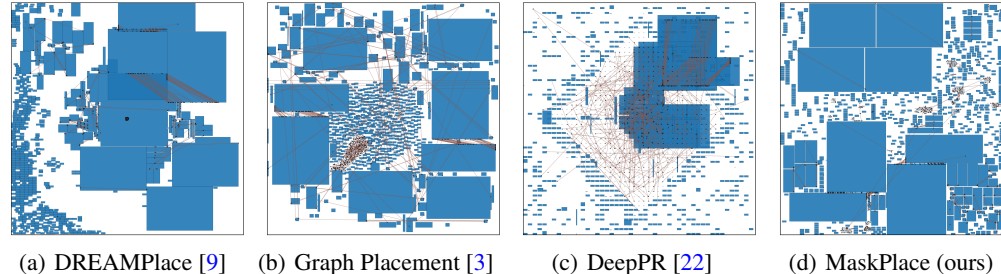

| (a) DREAMPlace [9] | (b) Graph Placement [3] | (c) DeepPR [22] | (d) MaskPlace (ours) |

Figure 1: **Visualizing different placements of a circuit benchmark** *bigblue3*, where the modules are visualized by blue rectangles and the wires are shown in brown lines to connect massive pins on modules. For clarity, we only show 1% wires. The proposed MaskPlace is compared with three representative approaches, including **(a)** DREAMPlace [9] (HPWL $= 1.04 \times 10^7$, WL $= 1.08 \times 10^7$, OL $= 8.06\%$), **(b)** Graph Placement [3] (HPWL $= 3.45 \times 10^7$, WL $= 3.73 \times 10^7$, OL $= 0.80\%$), **(c)** DeepPR [3] (HPWL $= 4.39 \times 10^7$, WL $= 5.18 \times 10^7$, OL $= 85.23\%$), and **(d) MaskPlace** (HPWL $= \underline{0.83 \times 10^7}$, WL $= \underline{0.88 \times 10^7}$, OL $= \underline{0\%}$), where HPWL, WL, and OL represent half-perimeter wirelength[2], wirelength, and overlap area ratio, respectively. All the metric values are smaller the better. The best performances are underlined in (d). We see that MaskPlace surpasses the recent popular placement approaches in all key metrics, and it can satisfy the $0\%$ hard density constraint. **Better zoom in 400%.**

examples of the performance metrics are given in Fig.8 and Fig.9 in Appendix. As pointed out in [3], the design space of placement is larger than $10^{2,500}$ when there are just $1,000$ circuit modules, whereas neural architecture search (NAS) typically has a space of $10^{30}$ and the Go game has a state space of $10^{360}$.

Methods of chip placement can be generally divided into two categories, classic optimization-based approaches [4–21] and learning-based approaches [3, 22, 23]. In the first category, hardware scientists often formulate placement as an optimization problem and relax the hard constraints. For example, let a pair of vectors $(\boldsymbol{x}, \boldsymbol{y})$ denote the $(x, y)$-coordinate value of all circuit modules on a 2D canvas, the objective function of placement can be formulated as minimizing $\mathrm{WL}(\boldsymbol{x}, \boldsymbol{y})$, subject to $\mathrm{D}(\boldsymbol{x}, \boldsymbol{y}) \leq \alpha$, where $\mathrm{WL}(\cdot, \cdot)$ and $\mathrm{D}(\cdot, \cdot)$ are the estimation functions of wirelength and density respectively, and $\mathrm{D}(\boldsymbol{x}, \boldsymbol{y}) \leq \alpha$ is a hard constraint with a very small density value $\alpha$, which ensures that all modules do not overlap. For instance, DREAMPlace [9] is a recent advanced method that minimizes $\mathrm{WL}(\boldsymbol{x}, \boldsymbol{y}) + \lambda \mathrm{D}(\boldsymbol{x}, \boldsymbol{y})$, which relaxes the hard density constraint. However, it cannot directly produce a valid and manufacturable layout because the non-overlapping constraint is not satisfied after relaxation. These approaches often need a post-processing step, such as manual refinement and legalization (LG), to remove the overlapping in placement, resulting in two issues, (1) the wirelength may increase substantially after LG, and (2) no feasible solution can be found if the available chip area is insufficient before post-processing.

In the second category, reinforcement learning (RL) is employed to solve placement as a sequential decision-making problem, placing each circuit module at a time. Although the learning-based approaches are still in their early stage, they can produce promising results to automate the chip design flow end-to-end significantly without human effort. For instance, Graph Placement [3] and DeepPR [22] represent a netlist as a hypergraph, denoted as $G = (V, E)$, where $V$ represents a set of nodes, and each node is a module, and $E$ is a set of edges, which are the wires connecting all modules. They train RL agents to place one module at a time by maximizing the metric values as rewards. However, the hypergraph is not scalable to comprehensively encode information of a netlist. For example, the relative positions (offsets) of pins are discarded in [3, 22]. The wirelength estimation is inaccurate without the pin information, but encoding this rich information would make the hypergraph too complicated because each module can have hundreds of pins. Furthermore, placement on a large hypergraph requires heavy computations. Mirhoseini et al. [3] reduced computations by placing 15% of the modules using reinforcement learning (the remaining modules are placed by classic method), and Cheng and Yan [22] decreased the size (resolution) of module and chip canvas as shown in Table 1. Both of them sacrificed their placement performance.

---

[2]HPWL (Half Perimeter Wire Length) is a common approximation metric of the wirelength and can be computed much more efficiently than wirelength.

Table 1: **Comparisons** of representative placement methods in different aspects, including method types ("Family"), canvas size ("Resolution"), state space, "0% overlap" (if the method can produce a layout without overlapping placement), training/inference speed ("Efficiency"), and the performance metrics to be optimized. We see that MaskPlace can outperform recent advanced methods by performing placement on a full canvas size of 224×224 (much larger than prior works) and producing a valid placement with 0% overlap (which cannot be achieved by previous methods). MaskPlace can also be trained and tested efficiently.

| | Family | Resolution | State Space | 0% Overlap | Reward | Efficiency | Metrics |
|---|---|---|---|---|---|---|---|
| DREAMPlace [9] | Nonlinear | Continuous | - | ✗[1] | - | - /High | H, D [2] |
| Graph Placement [3] | RL+Nonlinear | $128^2$ | $(128^2)^{\alpha V}$ [3] | ✗ | Sparse | Med./Med. | H, C, D |
| DeepPR [22] | RL | $32^2$ | $(32^2)^V$ | ✗ | Dense | High/Med. | H, C |
| MaskPlace (ours) | RL | $224^2$ | $(224^2)^V$ | ✔ | Dense | High/High | H, C, D |

[1] DreamPlace needs a post-processing step, such as legalization (LG) that may fail.

[2] H = HPWL, C = Congestion, D = Density.

[3] $V$ is the number of circuit modules and $\alpha \approx 15\%$ in Graph Placement.

To address the issues of prior arts, we propose a novel RL method, named MaskPlace, which can automatically generate a high-quality and valid layout (non-overlapping modules) within a few hours, unlike previous methods that need manual refinement to modify invalid placement, which may wait up to 72 hours for commercial electronic design automation (EDA) tools to evaluate the placement. MaskPlace casts placement as a problem of pixel-level visual representation learning for circuit modules using convolutional neural networks. This representation can comprehensively capture the configurations of thousands of pins, enabling fast placement in a full action space on a large canvas size *e.g.,* 224×224. As shown in Fig.1 and Table 1, MaskPlace has many attractive benefits that existing works do not have. MaskPlace is mainly for macro placement due to the problem size.

This paper has three main **contributions**. Firstly, we recast chip placement as a problem of learning visual representation to describe millions of circuit modules on a chip comprehensively. It opens up a new perspective for AI-assist chip placement. Secondly, we carefully design a new policy network that can capture and aggregate both the global and subtle information on a chip canvas, maximizing the reward of wirelength and ensuring non-overlapping placement efficiently. Thirdly, extensive experiments demonstrate that MaskPlace outperforms recent advanced methods on 24 public chip benchmarks. For example, MaskPlace can always produce a layout with 0% overlap while reducing wirelength up to 5× and 9× compared to Graph Placement [3] and DeepPR [22] respectively.

## 2  Preliminary and Notation

The placement quality can be measured by the HPWL (half perimeter wirelength), which estimates the wirelength with marginal computational cost [24]. Intuitively, Fig.2(e) illustrates a 2D chip canvas. Let $M^i$ and $P^{(i,j)}$ denote the $i$-th module and its $j$-th pin, respectively. A net contains a set of pins connecting modules by wires. For example, "Net 1" (in red) connects all four modules (*i.e.,* $M^1, M^2, M^3, M^4$) using wires through pins $P^{(1,2)}$, $P^{(2,2)}$, $P^{(3,2)}$, and $P^{(4,1)}$, while "Net 2" (in green) connects three modules (*i.e.,* $M^1, M^2, M^3$) using wires through pins $P^{(1,1)}$, $P^{(2,1)}$, and $P^{(3,1)}$. HPWL estimates the wirelength by summing up the half perimeters of bounding boxes of all the nets, as shown by the red and green boxes in Fig.2(e). Intuitively, the half perimeter of a net bounding box equals the sum of its height and width. For example, HPWL in Fig.2(e) is $h_1 + w_1 + h_2 + w_2$.

Given a netlist containing a set of nets, minimizing the wirelength can be treated as minimizing HPWL by placing modules to the optimal positions on a 2D chip canvas. To achieve a valid and manufacturable chip layout, we need to satisfy two hard constraints: (1) *congestion constraint*: the wire congestion should be lower than a desired small threshold to reduce chip cost, and (2) *overlap constraint*: the density should be minimized to achieve non-overlapping placement.

$$\min \sum_{\forall \text{net} \in \text{netlist}} \left( \max_{P^{(i,j)} \in \text{net}} P_x^{(i,j)} - \min_{P^{(i,j)} \in \text{net}} P_x^{(i,j)} + \max_{P^{(i,j)} \in \text{net}} P_y^{(i,j)} - \min_{P^{(i,j)} \in \text{net}} P_y^{(i,j)} \right)$$

$$\text{s.t.} \quad \text{Congestion}(M_x, M_y, M_w, M_h) \leq C_{\text{th}} \quad \text{and} \quad \text{Overlap}(M_x, M_y, M_w, M_h) = 0,$$

(1)

where $P_x$ and $P_y$ represent the $(x, y)$-coordinate value of a pin respectively, $\text{Congestion}(\cdot)$ is the congestion function, $C_{\text{th}}$ is a desired threshold, $\text{Overlap}(\cdot)$ is the overlap function, and $M_x, M_y, M_w, M_h$

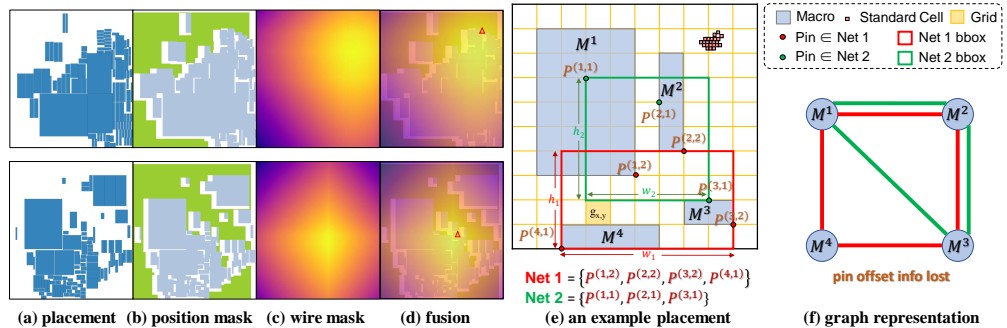

| (a) placement | (b) position mask | (c) wire mask | (d) fusion | (e) an example placement | (f) graph representation |

Figure 2: **Mask Visualization, placement example, and hypergraph representation in prior work.** We visualize different masks in MaskPlace (a-d) and illustrate an example of placement in (e). In the position mask (b), the green color means feasible positions to place while the gray color represents the placed modules. In the wire mask (c), lighter color indicates shorter wirelength if a module is placed at a specific position. The fusion mask in (d) is an example of the output after the mask fusion model using $1 \times 1$ convolutions, where the $\triangle$ denotes the position with a high probability to place at (*i.e.,* no overlap and shorter wirelength). (f) is the result when converting the circuit in (e) into a hypergraph in prior works, where the critical information of pin locations is lost.

represent the position, width, and height of modules respectively. Firstly, lower congestion often indicates shorter wirelength, which is crucial to reduce chip cost because the wire resources are limited on a real chip. Inspired by prior arts [3, 22], we employ the RUDY estimator [25] to estimate wire congestion. Details of RUDY can be found in the Appendix A.2. Secondly, the placement density calculates the overlapping region between every pair of circuit modules. It is time-consuming since its computational complexity is $\mathcal{O}(V^2)$ where $V$ is the number of modules [1]. The proposed approach can ensure non-overlapping placement to avoid calculating this density metric explicitly in training, thus reducing computations while producing a valid layout.

## 3 Our Approach

**Model Architecture Overview.** Chip placement can be formulated as a Markov Decision Process (MDP) [26] by placing each module at a time. Fig.4 illustrates the overall architecture of MaskPlace, which trains a policy $\pi_\theta(a_t|s_t)$ represented by a convolutional encoder-decoder network with parameter set $\theta$, and a value function $V_\phi(s_t)$ represented by an embedding model with parameter set $\phi$. The policy network receives previous observations and actions as input $s_t$ and selects an action $a_t$ as output. Specifically, $s_t$ is a set of pixel-level feature maps that comprehensively capture the net and pin configurations in $M^{1:t-1}$, $M^t$, and $M^{t+1}$, where $M^{1:t-1}$ denotes the modules that have been placed in the previous time steps from 1 to $t-1$, while $M^t$ and $M^{t+1}$ denote the modules to be placed at the current step $t$ and the next step $t+1$, respectively. Intuitively, MaskPlace looks one step forward to achieve better placement.

Although prior arts [3, 22] represented a netlist as a hypergraph as shown in Fig.2(f) where each node is a module, and each edge is a wire between two modules, they lost the information of pin offsets for each module. Unlike previous works, MaskPlace can fully represent massive net and pin configurations using three types of pixel-level feature maps, as shown in Fig.2(a-d), including position mask, wire mask, and view mask, as discussed below. Different masks are fused by convolutions to learn the state representation.

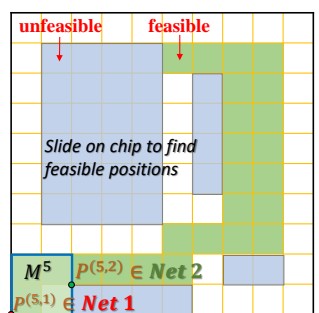

Figure 3: Position Mask Example.

**Position Mask.** The position mask, denoted by $f_p \in \{0,1\}^{224 \times 224}$, is a binary matrix of a canvas grid with size $224 \times 224$ as shown in Fig.3, where value "1" means a feasible position to place a module. The purpose of the position mask is to guarantee no overlaps between modules (*i.e.,* satisfy the overlap constraint) and to learn the relationship between placement and wirelength. Specifically, we slide a module $M^t$ (for example,

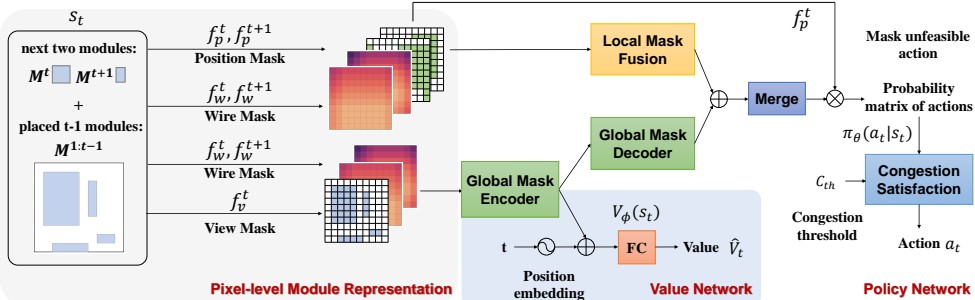

Figure 4: **Overview of MaskPlace,** which contains three main parts: a pixel mask generation model, a policy network, and a value network. The pixel mask generation model converts the current placement state into pixel-level masks. The policy and value networks convert these masks to actions and values based on global and local features. The congestion satisfaction block is to satisfy the congestion constraint and give the final action.

$t = 5$) on the entire chip canvas. The trajectory of the feasible positions (in green) can be labeled with "1". Intuitively, we can check each position for each module using the cumulative sum array [27]. This naive approach has the computational complexity of $\mathcal{O}(N^2)$ when a 2D canvas grid is divided into $N \times N$ cells. However, this simple approach is not efficient when $N$ is large. Therefore, since all modules are rectangles, we design an efficient generation algorithm, which iterates through all placed modules (in blue) and excludes positions that will cause overlap. In this case, all remaining positions are available for placement. The new algorithm is summarized in Appendix A.3, which costs $\mathcal{O}(V)$ for each module, where $V$ is the number of modules.

**Wire Mask.** The wire mask, denoted as $f_w \in [0,1]^{224 \times 224}$, is a continuous matrix for representing how HPWL increases if we place a module $M^t$ in a specific position. Fig.5 shows a sample of wire mask, where each value means the increase of HPWL. The wire mask aims at finding the best position with the minimum increase of the wirelength. Intuitively, we can calculate the HPWL at each canvas position, leading to a complexity of $\mathcal{O}(N^2P)$, where $P$ is the total number of pins. However, a fast algorithm can be designed by considering the relationships between the pin offset, the net bounding box, and the linear property of the HPWL metric. For example, Fig.3 illustrates that the next module $M^5$ has two pins, $P^{(5,1)}$ and $P^{(5,2)}$, belonging to "Net 1" and "Net 2" respectively (Fig.2(e)). Fig.5 illustrates the increase of wirelength when placing $M^5$ at each canvas location. For instance, if $M^5$ is at the bottom-left corner, its Manhattan distance to the two net

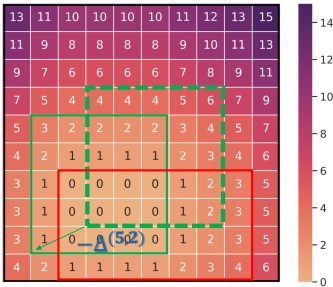

Figure 5: Wire Mask Example.

bounding boxes (in red and green) is $2 + 2 = 4$. To calculate the Manhattan distance more accurately, we move the net bounding box compared to the pin location. For example, since $P^{(5,2)}$ is located at $(2,1)^3$, we move the bounding box of Net 2 (in green) in the direction $-\Delta^{(5,2)} = (-2,-1)$ to encode the information of pin offset. The time complexity can be reduced to $\mathcal{O}(NP)$. The algorithm can be found in Appendix A.3.

**View Mask.** The view mask, denoted as $f_v \in \{0,1\}^{224 \times 224}$, is a global observation of the current chip layout, where the value "1" means a module has occupied this grid cell. Different from DeepPR [22] that assumed all modules have unit size, we consider real sizes of modules. For instance, if a module has size $w \times h$, it covers $\lceil wN/W \rceil \times \lceil hN/H \rceil$ cells in the canvas, where $W$ and $H$ represent the canvas size and $\lceil \cdot \rceil$ denotes the ceiling function.

**Learning Algorithm.** We train different blocks in Fig.4 as a whole using reinforcement learning. The detailed network architectures are provided in Appendix A.4. Firstly, we apply the above masks to represent the entire circuits and feed them to downstream networks. Secondly, a global feature encoder embeds the view mask of current placement and the wire masks of the following two steps into an embedding vector. Then we combine it with the positional embedding of the $t$-th circuit module in the value network to generate a scalar to evaluate the current state by fully-connected layers.

---

[3] We index the bottom-left corner as the origin $(0,0)$ in a two-dimensional coordinate.

Thirdly, a global mask decoder recovers a feature map of size $N^2$, which is fused with different position masks and wire masks in the policy network using $1 \times 1$ convolutions to avoid the local signal diffusion. The policy network predicts a probability action matrix of size $N \times N$, indicating where to put the next module. Before sampling actions, we can remove unfeasible actions using the position mask. Finally, the congestion satisfaction block applies the congestion threshold on the probability matrix to select a final action.

**Reinforcement Learning.** We borrow the representative actor-critic diagram [28] and PPO2 framework [29] to train the policy $\pi_\theta(a_t|s_t)$, where the state representation $s_t$ is listed in Table 11 in Appendix. The action $a_t$ is the canvas position (cell) to place the circuit module. Specifically, we treat the chip canvas as a grid and divide it into $N \times N$ cells, leading to $N^2$ possible actions. The objective function of the policy network can be formulated as

$$L_{\text{policy}}(\theta) = \hat{\mathbb{E}}\Big[ \min\big(r_t(\theta)\hat{A}_t,\ \text{clip}(\mathrm{r_t}(\theta), 1-\epsilon, 1+\epsilon)\hat{A}_t\big)\Big], \tag{2}$$

where the ratio $r_t(\theta) = \frac{\pi_\theta(a_t|s_t)}{\pi_{\theta_{\text{old}}}(a_t|s_t)}$ and $\hat{A}_t = G_t - \hat{V}_t$ denotes the advantage function. We employ $G_t = \sum_{k=0}^{V-t-1} \gamma^k r_{t+k+1}$ that is the cumulative discounted reward and $\hat{V}_t$ is the estimated value produced by the value network. We update the the value network by optimizing its objective, $L_{\text{value}}(\phi) = \hat{\mathbb{E}}\big[(G_t - \hat{V}_t)^2\big]$.

**Reward $r_t$.** We treat HPWL as the reward because wirelength is the main optimization target in different performance metrics. This is different from prior arts [3, 22] that weighted combines HPWL and congestion as the reward, which introduces the weighting coefficient as an extra hyper-parameter to tune. Specifically, we achieve a dense reward by defining a partial HPWL, which only computes the currently placed pins. For example, the partial HPWL for $t$ modules can be defined as $\text{HPWL}_t$. In other words, we compute $\text{HPWL}_t$ after taking action $a_t$. The reward for the step $t$ is $r_t = \text{HPWL}_{t-1} - \text{HPWL}_t$, *i.e.,* the opposite number of the increase of HPWL. Furthermore, instead of computing HPWL at each step, we can maintain the ranges of all net bounding boxes in one episode and update the changes of their sizes with a cost of $\mathcal{O}(P)$, where $P$ is the number of pins. Thus we can generate dense rewards while maintaining efficiency.

**Training and Testing.** Before training, we follow previous work [3] to sort the circuit modules according to the number of nets, areas, and the number of connected modules that have been placed to determine the place order. In training, we update the policy and value networks at each epoch by ignoring the congestion satisfaction block. When updating the value network, we stop the gradient back-propagated in the global mask encoder to avoid influence on the policy network. The detailed training setup is provided in Appendix A.5.

In the testing stage, for each step $t$, we obtain a probability matrix from the policy network and sample one place action $a_t$. Then, the congestion satisfaction block will check whether the congestion exceeds a threshold $C_{\text{th}}$ after applying this action. If it exceeds, we uniformly sample a few actions, look up the corresponding values from the wire mask $f_w^t$, and estimate the congestion before taking these actions. We choose the action with the minimal value in $f_w^t$ and the congestion lower than $C_{\text{th}}$. If all actions cannot satisfy congestion $C_{\text{th}}$, we select the action with the minimal congestion and move to the next step. Detailed congestion satisfaction algorithm can be seen in Appendix A.3.

## 4   Experiments

We extensively evaluate MaskPlace and compare it with several recent advanced placement methods, including NTUPlace3 [6], RePlAce [8], DREAMPlace [9], Graph Placement [3], and DeepPR [22], where NTUPlace3, RePlAce and DREAMPlace are optimization-based methods, whilst Graph Placement and DeepPR are learning-based approaches. All of them are evaluated on different public circuit benchmarks. All previous works are evaluated by following their experimental settings.

**Benchmark.** We evaluate MaskPlace in 24 circuit benchmarks selected from public datasets including the widely-used ISPD2005 [30], IBM benchmark suite [31], and Ariane RISC-V CPU design [32]. The number of evaluated benchmarks is three times more than previous work [9, 22, 3]. The statistics of benchmarks are given in Table 14 in Appendix A.6, where the largest circuit contains 1,293 macros, 22,802 pins, and more than a million standard cells, leading to a vast state space as aforementioned.

**Main Results.** Table 2 compares the HPWL results between all the above methods to place all macros. To enable a fair comparison, we evaluate all approaches[4] by using five random seeds and report the means and variances. Since the original DeepPR method did not capture macro size (thus does not avoid overlap between adjacent macros because all macros have unit size), we extend DeepPR to model macro size to reduce the overlapping ratio. We name it "DeepPR-no-overlap". Similar to prior works, we use the minimum spanning tree algorithm [33] to estimate routing wirelength [34]. From Table 2, we can see that MaskPlace achieves the lowest wirelength in six out of seven benchmarks (except "adaptec4" where it still outperforms all learning-centric methods). We also see that the conventional optimization-based approaches may fail when the circuit benchmark has high chip area usage, such as "bigblue3 " and "ariane". Also, MaskPlace gets the lowest wirelength compared with Graph Placement and simulated annealing [35] in the IBM benchmark, which is shown in Appendix A.7. This project website[5] visualizes and compares different placements.

Table 2: Comparisons of HPWL ($\times 10^5$). HPWL is the smaller the better. We see that MaskPlace outperforms other methods by large margins in six out of seven benchmarks. The traditional optimization such as NTUPlace3 and DREAMPlace may fail in a few benchmarks such as "ariane".

| Method | adaptec1 | adaptec2 | adaptec3 | adaptec4 | bigblue1 | bigblue3 | ariane |
|---|---|---|---|---|---|---|---|
| Random | 61.00±3.85 | 483.12±13.65 | 576.25±16.03 | 600.07±14.17 | 36.67±3.18 | 918.05±43.49 | 52.20±0.90 |
| NTUPlace3 [6] | 26.62 | 321.17 | 328.44 | 462.93 | 22.85 | 455.53 | LG fail |
| RePlAce [8] | 16.19±2.10 | 153.26±29.01 | 111.21±11.69 | 37.64±1.05 | 2.45±0.06 | 119.84±34.43 | LG fail |
| DREAMPlace [9] | 15.81±1.64 | 140.79±26.73[1] | 121.94±25.05 | **37.41±0.87** | 2.44±0.06 | 107.19±29.91[2] | LG fail |
| Graph Placement [3] | 30.10±2.98 | 351.71±38.20 | 358.18±13.95 | 151.42±9.72 | 10.58±1.29 | 357.48±47.83 | 16.89±0.60 |
| DeepPR [22] | 19.91±2.13 | 203.51±6.27 | 347.16±4.32 | 311.86±56.74 | 23.33±3.65 | 430.48±12.18 | 52.20±0.89 |
| DeepPR-no-overlap [22] | 47.39±4.02 | 425.86±19.59 | 545.40±16.40 | 525.51±10.85 | 26.29±1.48 | 815.10±40.36 | 62.82±0.82 |
| MaskPlace | **6.38±0.35** | **73.75±6.35** | **84.44±3.60** | 79.21±0.65 | **2.39±0.05** | **91.11±7.83** | **14.63±0.20** |

[1] 2 (of 5) seeds fail in legalization (LG).
[2] 1 (of 5) seed fails in legalization (LG).

**Compare to Graph Representation.** Since Graph Placement [3] is the recent advanced learning-based approach that employs hypergraph for placement, we compare MaskPlace with it in all four performance metrics, including HPWL, congestion, density, and overlap area ratio. Table 3 and 4 report the results. The overlap area ratio describes the ratio of the overlapping area between macros divided by the chip area. In Table 3, MaskPlace (soft constraint) means that the round function rather than the ceiling function is used to calculate the number of grid cells occupied by the placed macros. MaskPlace with soft constraints may produce better HPWL and congestion than its counterpart with hard constraints, but the overlap area ratio could not be zero because the constraints have been relaxed. From Table 3 and 4, we see that MaskPlace outperforms graph placement by large margins, especially in the ISPD benchmark, where the former reduces HPWL compared to the latter one by up to 80% while ensuring zero overlaps in all benchmarks. More results in the IBM benchmark can be found in Appendix Table 15.

Table 3: Comparisons between GraphPlace [3] and the proposed MaskPlace using different performance metrics (normalized to $[0, 1]$) in the "ariane" benchmark, including HPWL, congestion, density, and overlap area ratio. All values are smaller the better. We see that MaskPlace surpasses other methods significantly while ensuring zero overlaps, which is essential for a valid and manufacturable chip layout.

| Method | HPWL | Congestion | Density | Overlap (%) |
|---|---|---|---|---|
| Graph Placement (journal) [3] | 0.1198±0.0019 | 0.9718±0.0346 | 0.5729±0.0086 | 5.13±0.11 |
| Graph Placement (github) [3] | 0.1013±0.0036 | 0.9174±0.0647 | 0.5502±0.0568 | 4.29±1.25 |
| MaskPlace (hard constraint) | 0.1025±0.0015 | 1.0137±0.0451 | **0.5000±0.0000** | **0.00±0.00** |
| MaskPlace (soft constraint) | **0.0879±0.0012** | **0.9049±0.0115** | 0.5262±0.0015 | 3.33±0.79 |

**Routing Wirelength.** Table 5 compares the routing wirelength between MaskPlace and DeepPR [22]. We show that using the true wirelength as the reward would lower the efficiency and produce a sparse reward. We see that MaskPlace, which employs HPWL as the reward, can achieve 60% to 90% shorter routing wirelength than DeepPR, which directly used the true wirelength as the reward.

**Standard Cells.** Table 6 compares the HPWL of both the macros and the standard cells by using MaskPlace, DeepPR [22], and DREAMPlace [9], where DREAMPlace is employed to place the standard cells following the experimental setup in [22]. We can see that the proposed method

---

[4]The random seed does not apply in a classic method NTUPlace3.
[5]laiyao1.github.io/maskplace

Table 4: Comparisons between GraphPlace [3] and MaskPlace in four performance metrics (normalized to [0, 1]) in the ISPD benchmark. All values are smaller the better. We see that MaskPlace can reduce the HPWL up to 80% compared to its counterpart while ensuring the modules' overlaps are zeros in all benchmarks.

| benchmark | Graph Placement [3] | | | | MaskPlace | | | |
|---|---|---|---|---|---|---|---|---|
| | HPWL | Congestion | Density | Overlap(%) | HPWL | Congestion | Density | Overlap (%) |
| adaptec1 | 0.1810 | 0.7370 | 0.5340 | 1.89 | **0.0384** | **0.6961** | **0.5000** | **0.00** |
| adaptec2 | 0.2814 | 0.7387 | 0.5147 | 1.54 | **0.0549** | **0.6990** | **0.5000** | **0.00** |
| adaptec3 | 0.2248 | 0.7431 | 0.5226 | 1.24 | **0.0540** | **0.7130** | **0.5000** | **0.00** |
| adaptec4 | 0.1107 | 0.7369 | 0.7472 | 7.59 | **0.0560** | **0.7078** | **0.5000** | **0.00** |
| bigblue1 | 0.0958 | 0.7346 | 0.5181 | 1.98 | **0.0255** | **0.6953** | **0.4876** | **0.00** |
| bigblue3 | 0.1565 | 0.7499 | 0.5174 | 0.96 | **0.0430** | **0.7350** | **0.5000** | **0.00** |

Table 5: Compare routing wirelength ($\times 10^5$) between DeepPR [22] and MaskPlace.

| method | adaptec1 | adaptec2 | adaptec3 | adaptec4 | bigblue1 | bigblue3 | ariane |
|---|---|---|---|---|---|---|---|
| DeepPR [22] | 23.25±3.03 | 212.97±5.84 | 377.80±5.49 | 367.57±64.44 | 28.51±3.90 | 507.39±14.82 | 56.77±0.87 |
| DeepPR-no-overlap [22] | 52.46±3.97 | 451.22±19.00 | 583.32±15.92 | 628.22±10.02 | 31.02±1.41 | 945.60±43.24 | 68.89±0.81 |
| MaskPlace | **7.12±0.34** | **77.70±6.77** | **90.40±3.82** | **92.51±0.38** | **2.81±0.51** | **103.24±10.48** | **15.61±0.19** |

surpasses the other approaches by up to 50% in the large benchmark "bigblue3", which has more than a million standard cells.

Table 6: Comparisons of HPWL ($\times 10^7$) for macro and standard cell placement.

| Method | adaptec1 | adaptec2 | adaptec3 | adaptec4 | bigblue1 | bigblue3 |
|---|---|---|---|---|---|---|
| DREAMPlace [9] | 11.01±1.37 | 16.19±2.60 | 21.54±1.19 | 35.47±4.97 | 10.28±1.11 | 70.02±46.06 |
| DeepPR [22] + DREAMPlace [9] | 8.01 | 12.32 | 24.11 | 23.64 | 14.04 | 45.06 |
| MaskPlace + DREAMPlace [9] | **7.93±0.20** | **9.95±0.29** | **21.49±0.90** | **22.97±0.92** | **9.43±0.13** | **37.29±0.67** |

**Placement w/o real size**    Considering that DeepPR ignored the module size, we implement MaskPlace in the same setting, and the result can be found in Table 7. The result shows that our method has significant advantages.

Table 7: Routing wirelength for macro placement w/o real size

| Method | adaptec1 | adaptec2 | adaptec3 | adaptec4 | bigblue1 | bigblue3 |
|---|---|---|---|---|---|---|
| DeepPR [22] | 5298 | 22256 | 32839 | 63560 | 8602 | 94083 |
| MaskPlace | **2941** | **20593** | **16181** | **18553** | **2331** | **27403** |

**Transferability**    Test the performance of the model trained on *adaptec1* on other benchmarks as Table 8. The results show that our method also has a good transferability.

Table 8: HPWL ($\times 10^5$) results for transferability. HPWL is the smaller the better. The model has been trained on *adaptec1* benchmark and just took the inference in other benchmarks.

| | adaptec2 | adaptec3 | adaptec4 | bigblue1 | bigblue3 | ariane |
|---|---|---|---|---|---|---|
| HPWL | 85.56±9.41 | 89.77±6.72 | 87.32±3.93 | 2.87±0.31 | 160.63±10.41 | 19.32±2.02 |
| ratio[*] | 1.16 | 1.06 | 1.11 | 1.20 | 1.76 | 1.32 |

[*] Compared with the result from the model trained on the corresponding benchmark.

**Efficiency.** Table 9 compares the inference efficiency of different approaches. All of them are evaluated on one GeForce RTX 3090 GPU, and the CPU version of DREAMPlace is allocated with 16 threads in a 16 CPU cores environment. We see that the careful design of MaskPlace makes it faster than two other learning-based approaches.

**Ablation Study.** We compare different components in MaskPlace as shown in Fig.6. Each curve is produced by five seeds using the benchmark "adaptec1". For example, MaskPlace *w/ CL* means using 1/3 of the circuit macros to pretrain the model for 30 epochs like curriculum learning. MaskPlace *w/o* $M^{t+1}$ means only considering $M^t$ as input without looking one step forward. Moreover, MaskPlace

Table 9: Comparisons of wall-clock runtime (second) of different placement methods in inference.

| Method | adaptec1 | adaptec2 | adaptec3 | adaptec4 | bigblue1 | bigblue3 |
|---|---|---|---|---|---|---|
| DREAMPlace (CPU) [9] | 4.47 | 11.50 | 11.52 | 15.55 | 9.32 | 27.36 |
| DREAMPlace (GPU) [9] | 4.51 | 7.57 | 7.70 | **7.39** | 5.57 | **12.25** |
| Graph Placement [3] | 6.32 | 16.97 | 20.05 | 13.40 | 4.54 | 15.65 |
| DeepPR [22] | 10.25 | 10.46 | 22.82 | 42.24 | 9.86 | 32.53 |
| MaskPlace | **4.26** | **6.98** | **7.63** | 13.36 | **4.32** | 13.87 |

*w/o number of nets* means this feature is not considered when determining the placement order. MaskPlace *w/o 1x1 conv* means that 7x7 kernels are used to replace the 1x1 kernels in the local feature fusion block. Also, MaskPlace with *sparse reward* means compute HPWL reward only when all macros have been placed, and the rewards for other steps are set to zeros. MaskPlace *w/o view mask* and *w/o wire mask* means that the corresponding mask is not inputted into the model. We can see that MaskPlace (standard) with curriculum learning has the best performance.

**Congestion Satisfaction.** To evaluate our congestion satisfaction block, we implement a placement without any congestion threshold (*i.e.,* $\infty$) as shown in Fig.7. We evaluate the "adaptec3" benchmark, where MaskPlace outperforms DeepPR. We gradually lower the threshold $C_{th}$ from 60 to 10. We find that lower congestion leads to an increase in the HPWL. Our method can always satisfy the congestion constraint in five seeds in a suitable range (above 40 in this benchmark). If we continue to reduce the congestion threshold after a specific value (say 40 in Fig.7), we found that it hardly satisfies the threshold because nets must take up a certain amount of wire resources.

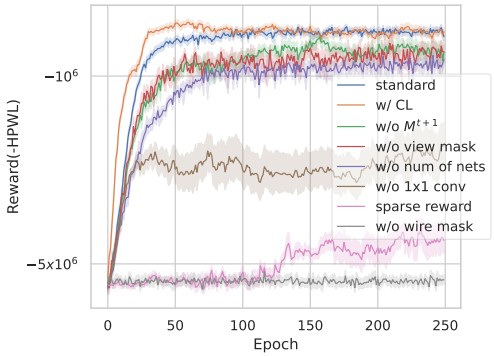

Figure 6: Compare reward curves of different components in MaskPlace.

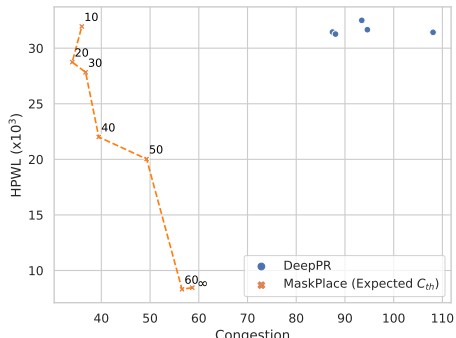

Figure 7: Study of congestion satisfaction.

## 5 Conclusion

This paper proposes MaskPlace, an RL-based placement method based on rich visual representation by learning position, wirelength, and view information. It helps the model take action effectively and efficiently without reducing the search space. We design a direct reward function based on practical scenarios and get satisfactory results on all key metrics. This work can facilitate the placement process and avoid undesired overlaps between modules. In the future, we will explore the standard cell placement by designing a suitable representation, which is an open problem for RL due to its vast space.

**Limitation and Potential Negative Societal Impact.** Chip design flow contains many stages, and our method shows its potential in a single stage. Similar to previous RL methods, it also requires an optimization method when placing millions of standard cells because RL's state space is too large. Our method does not have potential harm to the public society at the moment.

## Acknowledgments and Disclosure of Funding

We thank Xibo Sun from Huawei answering questions about EDA. We also thank Runjian Chen for participating in our discussion. Ping Luo is supported by the General Research Fund of HK No.27208720, No.17212120, and No.17200622.

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
