# A Appendix

## A.1 Module, Net and Pin

**Module.** A chip is a combination of numerous modules, and there are two types of them: macros and standard cells. Macros are relatively large, including DRAMs, caches, and IO interfaces. Standard cells are mainly logical gates, much smaller than macros, and the size can be ignored. As in Fig.8 (a), there are four macros and several standard cells. Placement methods usually place macros first and then the standard cells to ensure there is enough space for macros [36]. Due to the considerable number of standard cells, we currently use our MaskPlace method on macro placement.

**Pin.** Pins are input/output interfaces for modules and are connected by wires directly, which have fixed relative positions on modules. We define the relative position of the pin $P^{(i,j)}$ from the left-bottom corner of the module it belongs to as $\Delta^{(i,j)} = (\Delta_x^{(i,j)}, \Delta_y^{(i,j)})$. For example, there are five pins and three macros in Fig.9 (a), and the pin offset information is also shown at the bottom. In the placement task, we should not ignore the positions of pins because it determines the wirelength. However, graph neural network-based models [3, 22] ignored them when converting circuits into a graph, which may lead to sub-optimal results.

**Net.** A net contains a set of pins connected by the same wires. Thus the pins have the same information (0/1 in digital circuits). For example, four pins belong to Net 1, and the other three pins belong to Net 2 in Fig.8 (a). Usually, one pin belongs to only one net, and one net has more than two pins (one input and several outputs). Pins from the same net can form a net bounding box as Fig.8 (a)(b).

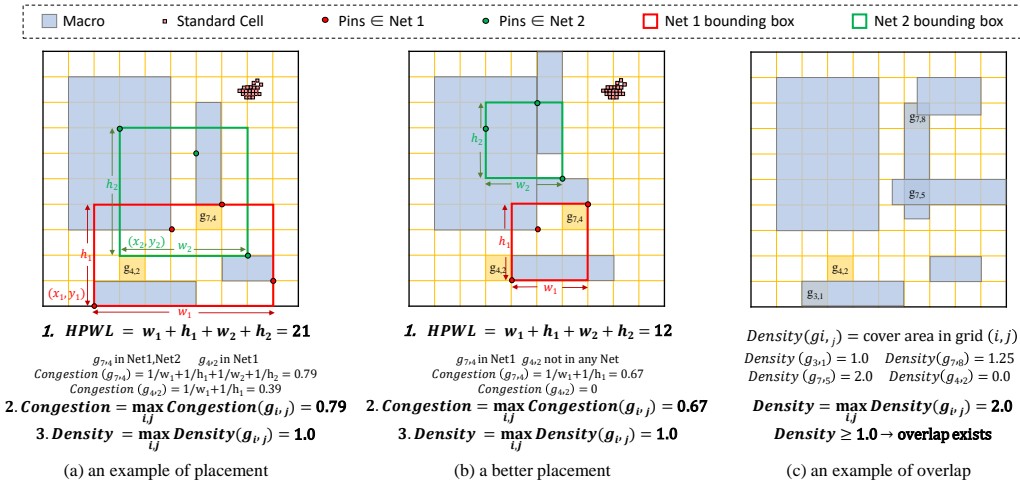

Figure 8: **Metrics for placement.** HPWL is an optimization item, while congestion and density are constraint items in the actual placement scenario. HPWL is smaller the better, while congestion and density need to be less than the given thresholds. Placement (b) is better than (a) because HPWL and congestion of (a) are smaller. Placement (c) is invalid because there are overlaps in cell $g_{7,5}$ and $g_{7,8}$.

## A.2 Metric

**HPWL.** HPWL (Half Perimeter Wire Length) is widely used to estimate wirelength by small computation cost [24]. It is the sum of half perimeter of net bounding boxes as Fig.8 (a)(b), where the bounding box is the minimal rectangle including all pins belonging to this net.

**Congestion.** The congestion metric is used to avoid routing congestion, resulting in an increase in the actual wirelength because the resources for wires are limited in a real chip. There are many ways to estimate congestion, one is to compute a rough routing result [3], but it is very computationally intensive. We use RUDY [25] as the estimation of congestion, which is a common way to evaluate. In RUDY, each grid cell needs to accumulate the inverse of the height and width $(1/h + 1/w)$ of all

the net bounding boxes covering itself and take out the maximum value (or the average of the first k maximums) of all grid cells as Fig.8 (a)(b).

**Density.**    Density is a metric to reduce overlaps and avoid time-consuming computation for $O(V^2)$ constraints [1]. So, it is an approximate calculation essentially. It is defined as the maximum stackable coverage area ratio for each grid cell on a chip canvas. For example, as Fig.8 (c), the maximum stackable coverage area ratio is 2.0 in grid cell $g_{7,5}$ because two modules fully occupy it. However, density less than a small value is not a sufficient condition for the absence of overlap. Because our method can ensure no overlaps, we only consider it in evaluation. In the practical application scenario of chip design, HPWL is an optimization item. Conversely, congestion and density are constraint items.

**Examples.**    We give a set of placement results to explain the metrics in Fig.8. We can see that HPWL is the sum of width and height of net bounding boxes. Congestion (RUDY) is the max congestion value of grid cell $g_{i,j}$, and the value in each grid cell is cumulative from the reciprocal of the width and height of the net bounding box containing that grid cell. (a) and (b) are from the same circuit, but (b) is a better placement because (b) has lower HPWL and congestion. Density is the max density value of grid cell $g_{i,j}$, and the value in each grid cell is stackable coverage area ratio of the grid cell. The density of Fig.8 (c) is 2.0 because $g_{7,5}$ completely covered by two modules.

**Relationship between pin offset and HPWL.**    The pin offset can affect the HPWL. In the graph-based method, the input features for module include size $(M_w, M_h)$, position $(M_x, M_y)$ and type. So, the network can hardly infer the real position of pins and tend to use the center positions of modules to predict the positions of pins. In this way, the agent will align the centers of the two modules horizontally, and the result of placement is like Fig.9 (b) to get the wirelength 6. However, when considering the pins are near the bottom of the modules, it is better to align the bottom of the two modules as Fig.9 (c), and thus wirelength can be reduced to 2 if we consider the pin offset.

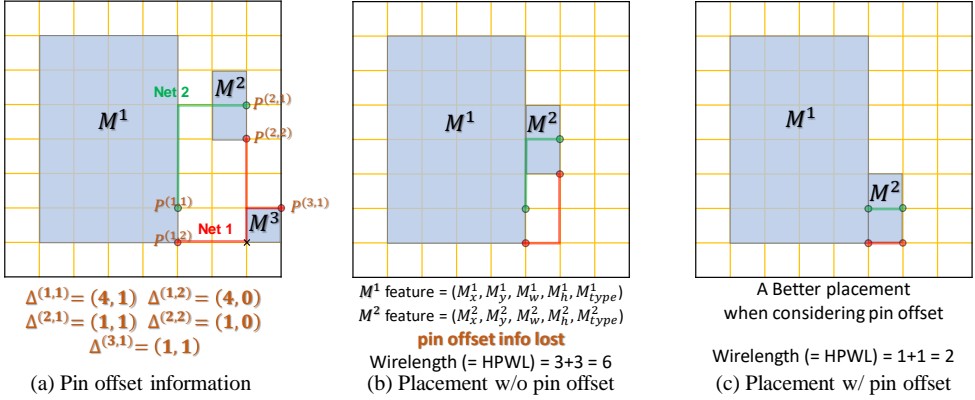

Figure 9: **Explanation for module, pin and net.** (a) gives an example for pin offset information. When we remove the pin offset information, the model tends to align the centers of the two modules horizontally like (b) because it uses the center position of modules to estimate pin location. However, we have a better design as (c) when considering pins are located on the bottom of the modules.

## A.3   Algorithms

**Reward Computation.**    The dense reward generation algorithm is shown in Algorithm 1. It can generate dense rewards without an efficiency decrease. For simplicity, we omit the calculation of the y dimension, which is the same as the x dimension.

**Position Mask Generation.**    The efficient position mask generation algorithm is in Algorithm 2.

**Wire Mask Generation.**    The efficient wire mask generation algorithm is shown in Algorithm 3. For simplicity, we omit the calculation of the y dimension, which is the same as the x dimension.

**Algorithm 1:** Dense HPWL Reward Computation (omit y-dimension)

---

**Data:** Placed position of module $M^t$ $(M_x^t, M_y^t)$, max/min x/y coordinates of nets $MaxMinCoord$, pin offsets $(\Delta_x^{(t,j)}, \Delta_y^{(t,j)})$, pin to net connection $P_n^{(t,j)}$;

**Result:** Incremental HPWL Reward $reward$;

$reward \leftarrow 0$;

**foreach** $\Delta_x^{(t,j)}, P_n^{(t,j)}$ *of all pins* $P^{(t,j)}$ *from* $M^t$ **do**

    $x \leftarrow M_x^t + \Delta_x^{(t,j)}$ ; // `calculate pin coordinates`

    **if** $P_n^{(t,j)}$ *not in* $MaxMinCoord$ **then**

        // `The net for the first time has a definite location of the pin`

        $MaxMinCoord[P_n^{(t,j)}].x.max \leftarrow x$;

        $MaxMinCoord[P_n^{(t,j)}].x.min \leftarrow x$;

    **else**

        // `Update the bounding box range`

        **if** $MaxMinCoord[P_n^{(t,j)}].x.max < x$ **then**

            $reward \leftarrow reward + (x - MaxMinCoord[P_n^{(t,j)}].x.max)$;

            $MaxMinCoord[P_n^{(t,j)}].x.max = x$;

        **else if** $MaxMinCoord[P_n^{(t,j)}].x.min > x$ **then**

            $reward \leftarrow reward + (MaxMinCoord[P_n^{(t,j)}].x.min - x)$ ;

            $MaxMinCoord[P_n^{(t,j)}].x.min = x$;

        **end**

    **end**

**end**

---

**Algorithm 2:** Position Mask Generation

---

**Data:** Width, Height and Position of t-1 placed module $M^{1:t-1}$ $(M_w^{1:t-1}, M_h^{1:t-1}, M_x^{1:t-1}, M_y^{1:t-1})$

**Result:** Position Mask $f_p^t$ for Module $M^t$

$f_p^t \leftarrow ones(N, N)$; // *$ones(N, N)$* `is all-ones` $N \times N$ `matrix`

**for** $i \leftarrow 1$ **to** $t - 1$ **do**

    $tmp \leftarrow ones(N, N)$;

    // `find positions that will cause` $M^t$ `and` $M^i$ `to overlap`

    $tmp[M_x^i - M_w^t + 1 : M_x^i + M_w^i - 1, M_y^i - M_h^t + 1 : M_y^i + M_h^i - 1] \leftarrow 0$;

    // `exclude infeasible positions`

    $f_p^t \leftarrow tmp \odot f_p^t$ ; // $\odot$ `is element-wise product`

**end**

---

---

**Algorithm 3:** Wire Mask Generation (omit y-dimension)

---

**Data:** Hash Map of Max/Min X/Y coordinates of nets $MaxMinCoord$, pin's offsets
$(\Delta_x^{(t,j)}, \Delta_y^{(t,j)})$, pin to net connection $P_n^{(t,j)}$
**Result:** Wire Mask $f_w^t$ for module $M^t$
$f_w^t \leftarrow zeros(N, N)$;
// Accumulate the wirelength increase for each net
**foreach** $\Delta_x^{(t,j)}, P_n^{(t,j)}$ *of all pins* $P^{(t,j)}$ *from* $M^t$ **do**
   // If the pin is to the left of the net bounding box
   **for** $i \leftarrow 0$ **to** $MaxMinCoord[P_n^{(t,j)}].x.min + \Delta_x^{(t,j)} - 1$ **do**
      $f_w^t[i, :] \leftarrow f_w^t[i, :] + MaxMinCoord[P_n^{(t,j)}].x.min + \Delta_x^{(t,j)} - i$;
   **end**
   // If the pin is to the right of the net bounding box
   **for** $i \leftarrow MaxMinCoord[P_n^{(t,j)}].x.max + \Delta_x^{(t,j)} + 1$ **to** $N - 1$ **do**
      $f_w^t[i, :] \leftarrow f_w^t[i, :] + i - (MaxMinCoord[P_n^{(t,j)}].x.max + \Delta_x^{(t,j)})$;
   **end**
**end**

---

**Congestion Satisfaction.** The algorithm implemented in the congestion satisfaction block can be seen in Algorithm 4.

---

**Algorithm 4:** Placement with Congestion Constraint

---

**Data:** Trained place agent $agent$, expected congestion threshold $C_{th}$
**Result:** A placement plan $[a_1, a_2, ..., a_V]$ that meet the congestion requirement
**for** $i \leftarrow 1$ **to** $V$ **do**
   Choose $a_i$ from the probability matrix generated by policy network $agent$;
   $Cong \leftarrow$ congestion matrix from the state after taking $a_i$;
   Compute congestion value $c$ from $Cong$;
   **if** $c > C_{th}$ **then**
      Randomly sample $N$ different actions $a_i^{1:N}$ from action space;
      Compute $N$ congestion values $c_i^{1:N}$ from congestion metrics;
      Get $N$ wirelength values $w_i^{1:N}$ from wire masks;
      Sort the $N$ actions according to $w_i^{1:N}$ (as the 1st key) and $c_i^{1:N}$ (as the 2nd key);
      $flag \leftarrow False$;
      **for** $j \leftarrow 1$ **to** $N$ **do**
         **if** $c_i^j \leq C_{th}$ **then**
            $flag \leftarrow True$;
            $a_i \leftarrow a_i^j$;
            break;
         **end**
      **end**
      // If all sampled actions cannot satisfy congestion threshold, we choose the one with minimal congestion increase.
      **if** $flag$ *is* $False$ **then** $a_i \leftarrow$ the action $a_i^j$ with minimum $c_i^j$;
   **end**
   Take action $a_i$ as the final action;
**end**

---

## A.4 Details of Model Architecture

The parameters of layers in model architecture are in Table 10. Also, the features used by pixel-level mask generation are in Table 11. The comparison of features for the placement order in different methods can be seen in Table 12.

Table 10: Model Architecture

| Block | Layer | Kernel Size | Output shape |
|---|---|---|---|
| Local Mask Fusion | Conv | $1 \times 1$ | $(224, 224, 8)$ |
| | Conv | $1 \times 1$ | $(224, 224, 8)$ |
| | Conv | $1 \times 1$ | $(224, 224, 1)$ |
| Global Mask Encoder | ResNet-18 | - | 1000 |
| | FC | - | 768 |
| Global Mask Decoder | Deconv | $3 \times 3$ | $(14, 14, 8)$ |
| | Deconv | $3 \times 3$ | $(28, 28, 4)$ |
| | Deconv | $3 \times 3$ | $(56, 56, 2)$ |
| | Deconv | $3 \times 3$ | $(112, 112, 1)$ |
| | Deconv | $3 \times 3$ | $(224, 224, 1)$ |
| Merge | Conv | $1 \times 1$ | $(224, 224, 1)$ |
| Position Embedding | - | - | 64 |
| FC for Value | FC | - | 512 |
| | FC | - | 64 |
| | FC | - | 1 |

Table 11: State Features

| Module status | Index | Feature | Notation | Dimension per module |
|---|---|---|---|---|
| Placed | $M^{1:t-1}$ | Width | $M_w$ | 1 |
| | | Height | $M_h$ | 1 |
| | | Position | $M_x, M_y$ | 2 |
| | | Pin Offset | $\Delta_x, \Delta_y$ | $2 \times$ num of pins |
| | | Pin to Net Connection | $P_n$ | num of pins |
| Unplaced | $M^t, M^{t+1}$ | Width | $M_w$ | 1 |
| | | Height | $M_h$ | 1 |
| | | Pin Offset | $\Delta_x, \Delta_y$ | $2 \times$ num of pins |
| | | Pin to Net Connection | $P_n$ | num of pins |

Table 12: Features used for placement order

| Method | Features for place order |
|---|---|
| Graph Placement [3] | Topological order, Area |
| DeepPR [22] | None |
| MaskPlace | Number of nets, Area, Number of its connected modules have been placed |

## A.5 Training Configuration

The detailed configuration and hyperparameter settings of our model is in Table 13.

Table 13: Model Configuration

| Configuration | Value | Configuration | Value |
|---|---|---|---|
| Optimizer | Adam | Learning rate | $2.5 \times 10^{-3}$ |
| Total epoch | 150 | Epoch for update | 10 |
| Batch size | 64 | Buffer capacity | $10 \times$ num of modules |
| Clip $\epsilon$ | 0.2 | Clip gradient norm | 0.5 |
| Reward discount $\gamma$ | 0.95 | Num GPUs | 1 |
| CPU | AMD Ryzen 9 5950X | GPU | GeForce RTX 3090 |

Also, we implement DREAMPlace[6] [9], Graph Placement[7] [3] ,and DeepPR[8] [22] by their open source codes with their default settings.

## A.6 Details of Benchmark

The detailed statistics of benchmarks are in Table 14. Hard macros are macros placed by the RL method in Graph Placement [3], and the remaining macros, also named soft macros, are placed by the classic optimization-based method. However, this distinction does not apply to the process of our method, which means we place all macros by RL. The statistical range of nets, pins, and area utilization are macros. Ports are terminals connecting to an external circuit, seen as fixed and no-size modules. Our method is also applicable to circuits with ports without additional modifications.

Table 14: Statistics of different chip benchmarks.

| Benchmark | Macros | Hard Macros | Standard Cells | Nets | Pins | Ports | Area Util(%) |
|---|---|---|---|---|---|---|---|
| adaptec1 | 543 | 63 | 210,904 | 3,709 | 4,768 | 0 | 55.62 |
| adaptec2 | 566 | 190 | 254,457 | 4,346 | 10,663 | 0 | 74.46 |
| adaptec3 | 723 | 201 | 450,927 | 6,252 | 11,521 | 0 | 61.51 |
| adaptec4 | 1,329 | 92 | 494,716 | 5,939 | 13,720 | 0 | 48.62 |
| bigblue1 | 560 | 32 | 277,604 | 657 | 1,897 | 0 | 31.58 |
| bigblue3 | 1,293 | 138 | 1,095,519 | 5,537 | 15,225 | 0 | 66.81 |
| ariane | 932 | 134 | 0 | 12,404 | 22,802 | 1,231 | 78.39 |
| ibm01 | 246 | 246 | 12,506 | 908 | 1,928 | 246 | 61.94 |
| ibm02 | 280 | 272 | 19,321 | 602 | 1,466 | 259 | 64.63 |
| ibm03 | 290 | 290 | 22,846 | 614 | 1,237 | 283 | 57.97 |
| ibm04 | 608 | 296 | 26,899 | 1,512 | 3,167 | 287 | 54.88 |
| ibm06 | 178 | 178 | 32,320 | 83 | 175 | 166 | 54.77 |
| ibm07 | 507 | 292 | 45,419 | 2,471 | 5,992 | 287 | 46.03 |
| ibm08 | 309 | 302 | 51,000 | 1,725 | 3,721 | 286 | 47.13 |
| ibm09 | 253 | 56 | 53,142 | 446 | 898 | 285 | 44.52 |
| ibm10 | 786 | 56 | 68,643 | 2,160 | 4,720 | 744 | 61.40 |
| ibm11 | 373 | 56 | 70,185 | 682 | 1,371 | 406 | 41.40 |
| ibm12 | 651 | 205 | 70,425 | 1,589 | 3,468 | 637 | 53.85 |
| ibm13 | 424 | 100 | 83,775 | 804 | 1,669 | 490 | 39.43 |
| ibm14 | 614 | 91 | 146,991 | 1,620 | 3,960 | 517 | 22.49 |
| ibm15 | 393 | 22 | 161,177 | 748 | 1,521 | 383 | 28.89 |
| ibm16 | 458 | 37 | 183,026 | 1,755 | 3,981 | 504 | 39.46 |
| ibm17 | 760 | 107 | 184,735 | 2,055 | 4,366 | 743 | 19.11 |
| ibm18 | 285 | 285 | 210,328 | 727 | 1,600 | 272 | 11.09 |

## A.7 Supplementary Experiment

**More benchmarks** We also conducted experiments in the IBM benchmark suite (ICCAD 2004) [31], which has been used to evaluate placement for more than a decade. We remove the "ibm05" because it does not contain any macros. We use our MaskPlace to place large macros and DREAMPlace

---

[6] github.com/limbo018/DREAMPlace
[7] github.com/google-research/circuit_training
[8] github.com/Thinklab-SJTU/EDA-AI

[9] to place standard cells. We compared our method with graph placement [3] and the simulated annealing method used in [3]. The results are in Table 15 and our method can achieve the lowest HPWL in all benchmarks.

Table 15: Comparisons of HPWL ($\times 10^5$) for macro and standard cell placement in ibm benchmark.

| Method | ibm01 | ibm02 | ibm03 | ibm04 | ibm05 | ibm06 |
|---|---|---|---|---|---|---|
| Graph Placement [3] | 31.71 | 55.12 | 80.00 | 86.86 | - | 63.48 |
| Simulated Annealing [3] | 25.85 | 54.87 | 80.68 | 83.32 | - | 69.09 |
| MaskPlace+DREAMPlace [9] | **24.18** | **47.45** | **71.37** | **78.76** | - | **55.70** |
| Method | ibm07 | ibm08 | ibm09 | ibm10 | ibm11 | ibm12 |
| Graph Placement [3] | 117.71 | 134.77 | 148.74 | 440.78 | 218.73 | 438.57 |
| Simulated Annealing [3] | 117.71 | 144.89 | 141.67 | 463.04 | 228.79 | 435.77 |
| MaskPlace+DREAMPlace [9] | **95.27** | **120.64** | **122.91** | **367.55** | **202.23** | **397.25** |
| Method | ibm13 | ibm14 | ibm15 | ibm16 | ibm17 | ibm18 |
| Graph Placement [3] | 278.93 | 455.31 | 520.06 | 642.08 | 814.37 | 450.67 |
| Simulated Annealing [3] | 259.89 | 405.80 | 510.06 | 614.54 | 720.40 | 442.00 |
| MaskPlace+DREAMPlace [9] | **246.49** | **302.67** | **457.86** | **584.67** | **643.75** | **398.83** |

For the larger circuit *bigblue4* in ISPD 2005 benchmark, the result of our method and baselines can been seen as Table 16. MaskPlace still achieved the best performance.

Table 16: HPWL ($\times 10^7$) results for *bigblue4* benchmark

| Benchmark | Random | NTUPlace3[6] | RePlAce[8] | DREAMPlace [9] |
|---|---|---|---|---|
| bigblue4 | 128.06±3.94 | 48.38 | 11.80±0.73 | 12.29±1.64 |
| Benchmark | Graph Placement [3] | DeepPR [22] | DeepPR-no-overlap [8] | MaskPlace |
| bigblue4 | 53.35±4.06 | 68.30±4.44 | 115.08±2.29 | **11.07±0.90** |

**Search time** We compared the search time of our method, Graph Placement [3] and DeepPR [22]. We tested all methods in the same environment and took the HPWL as the metric in benchmark *adaptec1*. The result is in Fig. 10. We can see that our approach can achieve the best performance in a few hours.

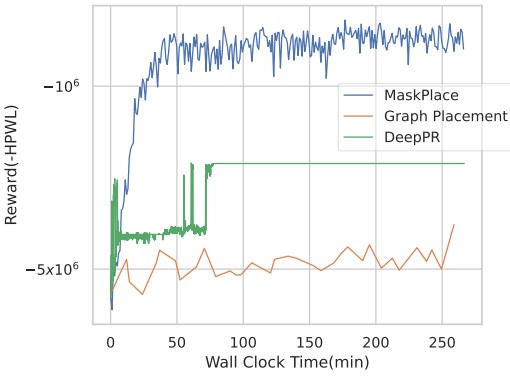

Figure 10: Search time comparison

## A.8 Detailed equation description of the model

We describe our model architecture in Fig. 4 in the form of equation.

With current state $s_t$, we first calculate the position masks $f_p^t, f_p^{t+1}$, wire masks $f_w^t, f_w^{t+1}$ and view mask $f_v^t$ via the mask generation function $m(\cdot)$.

$$f_p^t, f_p^{t+1}, f_w^t, f_w^{t+1}, f_v^t = m(s_t) \tag{3}$$

Then we extract the local feature $z_t^l$ via local mask fusion $g_\omega(\cdot)$ and the global features $z_t^g$ via the global mask encoder $enc_\eta(\cdot)$.

$$z_t^l = g_\omega(f_p^t, f_p^{t+1}, f_w^t, f_w^{t+1}) \tag{4}$$

where $g_\omega(\cdot)$ is a 1×1 convolutional neural network with parameter $\omega$.

$$z_t^g = enc_\eta(f_w^t, f_w^{t+1}, f_v^t) \tag{5}$$

where $enc_\eta(\cdot)$ is a convolutional neural network with ResNet-18 architecture with parameter $\eta$.

With local features $z_t^l$ and global features $z_t^g$, the state value $\hat{V}_t$ is derived by

$$\hat{V}_t = v_\phi(pos(t), z_t^g) \tag{6}$$

where $v_\phi$ is an MLP-like neural network with parameter $\phi$ and $pos(t)$ is an embedding vector which is related to step $t$.

We decode the global features $z_t^g$ into the dimension as same as the action space by the global mask decoder

$$z_t^{'g} = dec_\delta(z_t^g) \tag{7}$$

where $dec_\delta(\cdot)$ is a transpose convolutional neural network with parameter $\delta$.

Finally, we concatenate the local features $z_t^l$ and global features $z_t^{'g}$ in the channel dimension and merge them by another 1×1 convolutional neural network $\psi_\xi(\cdot)$. We further combine it with the position mask $f_p^t$ to generate action $a_t$ via the policy network $\pi_\theta(\cdot)$

$$a_t \sim \pi_\theta(\psi_\xi(z_t^l || z_t^{'g}), f_p^t) \tag{8}$$

where $\pi_\theta(\cdot)$ is an MLP-like neural network with parameter $\theta$ and $\psi_\xi(\cdot)$ is a 1×1 convolutional neural network with parameter $\xi$.

# B   Related Work

**Classic optimization-based methods.**   Optimization has been the dominant method in placement for decades. They can be divide into three categories: partitioning-based methods [4, 5], simulated annealing methods [10, 11] and analytical methods [6–9, 12–21].

Partitioning-based methods [4, 5] cluster the whole circuits into several parts to minimize the connections between parts. These methods first solve the placement problems within the same part and then place these parts to suitable positions on the chip based on the divide-and-conquer idea. However, optimizing modules within one part is an isolated problem, and sometimes it is hard to divide the circuits into relatively independent parts, which is highly related to the topology of the circuits.

Simulated Annealing (SA) methods [10, 11] are also known as hill-climbing methods, a widely used iterative heuristic algorithm for solving combinatorial optimization problems. They initialize a random status and then search for the following status by moving from the current status to a neighbor status. If the metrics of the neighbor status are better than that of the current status, they move to the neighbor status. Otherwise, the move may still be taken with a decreasing probability over time. The advantage is that they can be implemented when metrics do not have the analysis formula or cannot be differentiable. However, it is not efficient enough, and the placement results are highly dependent on the random initial state.

Analytical methods gradually replace the above two methods because of the best performance. They can be divided into quadratic methods [12–18] and nonlinear (non-quadratic) methods [6–9, 19–21]. Quadratic methods [12–18] transform the placement problem into a sequence of convex quadratic problems, and there are well-established solvers for such problems. However, it is a very rough approximation. Nonlinear methods [6–9, 19–21] design a single differentiable objective function and optimize it. The advantage is that it can handle large-scale modules. However, the objective function is still approximated, and they cannot avoid overlaps when combining multiple metrics in one objective function. Methods in this category achieved the highest placement quality among all classic methods [9].

**Learning-based methods.** With the development of deep learning, some learning-based approaches [11, 37–39] have been proposed to assist classic methods. Huang et al. [37] uses convolutional neural networks to estimate the congestion for SA placement. Vashisht et al. [11] uses the reinforcement learning models to generate the initial placement of SA. Kirby et al. [38], Agnesina et al. [39] help classic placement tools choose the most suitable hyperparameters with reinforcement learning methods. However, these methods do not implement end-to-end placement by deep learning, so the placement results depend heavily on the classic methods.

Pure reinforcement learning methods [3, 22, 23, 40] view placement as a process of placing modules sequentially. Mirhoseini et al. [3] uses reinforcement learning to place hard macros, and the force-directed method [18] to place remaining soft macros. Jiang et al. [23] replaces the force-directed method with DREAMPlace [9] to place soft macros based on Graph Placement [3]. Cheng and Yan [22] proposes a reinforcement learning method by using wirelength as the reward. Moreover, Chang et al. [40] puts all metrics in the RL reward. They have in common that they convert the circuit as a graph structure and input them to the graph neural networks [41]. However, the pin information has been lost, leading to sub-optimal placement. Also, they cannot avoid overlaps because of the reduction in search space. These methods still have room for improvement in terms of realistic chip placement. For instance, DeepPR [22] ignores the realistic size of the module. However, the size of the modules varies widely in most circuits. Although it proposes to use routing wirelength instead of HPWL as the reward, it will affect the efficiency and lead to sparse reward, making models hard to train. In contrast, HPWL is a high-quality wirelength estimation, and we do not need to discard this inherent dense reward.