# OpenReview forum: "MaskPlace: Fast Chip Placement via Reinforced Visual Representation Learning"
_NeurIPS.cc/2022/Conference — NeurIPS 2022 Accept_

### Official Review · Reviewer_iBtL · 2022-07-09

**Rating:** 5
**Confidence:** 5
**Soundness:** 2 fair
**Presentation:** 3 good
**Contribution:** 2 fair

**Summary:**

This paper proposes MaskPlace, which can automatically generate a high-quality and valid layout within a few hours. Unlike previous methods, that requires manual refinement to modify invalid placement, MaskPlace casts placement as a problem of pixel-level visual representation learning. More particularly, 1) Visual representation learning enables describing millions of circuit modules on a chip comprehensively. 2) MaskPlace suggests a new policy network that can capture and aggregate both the global and subtle information on a chip canvas, ensuring non-overlapping placement efficiently. 2) The demonstrated performance on 24 public chip benchmarks by outperforming Graph Placement and DeepPR 5x and 9x in reducing wirelength with 0% overlap in layout.

**Questions:**

1. The reviewer likes the paper for  reasons listed in above session. However, the reviewer is not quite sure about the adds-on values of thsi work compared to the original chip placement paper from Google [1] and why it is so important to the NeurIPS community. I would think the content in the paper makes it more suited for a design automation conference. The contributions in representation learning or algorithm design are small.

2. Can the authors provide some concrete comparisons on search time or optimization time with baseline algorithms? A great thing to have is to create a figure like Figure 6 but with different curves being the baselines. Make sure to plot wall clock time in the x-axis, instead of epochs as you might create a very complex network which can be 10x slower in step time, compared to the very simple baselines.

[1] "Chip Placement with Deep Reinforcement Learning", https://arxiv.org/abs/2004.10746

**Limitations:**

Yes.

**Strengths And Weaknesses:**

Strengths:
- The paper suggested a more accurate representation of chip layout: the three types of pixel-level feature maps can fully represent massive net and pin configurations. Different from DeepPR, that assumed all modules have unit size, this work considers real sizes of modules.

- The dense reward is an interesting idea . It is an inexpensive way to create an advantage value at each reinforce step, instead of generating one reward at only the end of the episode after the placement is done. This can be both good and bad those, as the decisions can be short-sighted such that the algorithm tries to incrementally improve wire length, losing performance opportunity from sacrificing short-term objectives. However, both the main results and ablation show that dense reward positively improves learning.

- The paper has thorough comparisons with SoTA related work and demonstrated significant gains on a set of benchmark. Metrics like wire length, density, and overlap are improved.

- The reviewer is happy to see Ablation Figure 6 of a decomposition of the gains. All the design decisions add up to some of the performance.

Weaknesses:
- The results look very strong, however, are a bit counter intuitive. The reward only considers wire length by continuously measuring an advantage function at each reinforce step. The reviewer is not so sure how it could improve density, which is demonstrated to be improved in all the results. Wire length and density are non-orthogonal metrics and are usually in conflict with each other.

- The paper lacks more details on how the baseline methods are implemented and whether the comparisons are fair and all the promising numbers are trustable.

- There is little formulations. Local mask fusion, global mask encoder and decoder, and other components in Figure 3 should be formulated mathematically for others to reproduce the experiments.

---

> ### Author Response · Authors · 2022-08-02
> **Response to Reviewer #iBtL (Part 2/2)**
>
> About questions:
>
> **Question 1 [why is it important to the NeurIPS community and the  contributions in representation learning or algorithm design are small]**
>
> Answer: Thanks a lot for your question. Our method does not incremental work on Graph Placement. We use an entirely different architecture.
>
> - **From the representation**, Graph Placement used the GCN-based model, which lost the pin offset information and made it difficult to learn the relationship between the graph structure. On the contrary, we use the pixel level masks as the placement representation and the mature CNN model to train. The proposed masks can provide implied information for placement, such as the position and wirelength, which can help the model learn better about chip placement.
>
> - **From the algorithm design**, we also proposed many beneficial designs for chip placement. These designs have been proven helpful in our ablation study.
>
>     - The local mask fusion module can help the model keep the position information by small convolutional kernel. Conversely, this information will be diffused when using large kernels.
>     - The design of dense rewards can make the reinforcement learning task easier to learn. The short-sighted decisions can be alleviated by adjusting the reward discount factor $\gamma$ to consider the rewards of multiple steps.
>     - Looking one more step forward can help the agent consider the relationship between the next two macros to place. Conversely, if we only consider the next step, the agent will put the next module into one position, which may prevent the following module from being placed in its desired position.
>     - Features used for ordering can help the agent place critical modules first when there are more available positions.
>     - Curriculum learning can accelerate and make the training process easier by training a model on the part of the circuit first.
>     - The generation algorithms for masks have low complexity by exploiting the properties of masks and can further speed up our approach.
>
> Also, the DeepPR paper [7] is published on NeuraIPS, which is also for chip placement tasks. We want to draw more attention to the task of the chip design in the AI community.
>
> We believe AI technology can significantly advance chip design technology in the future. Therefore our paper can help AI researchers to understand this field better and thus help them to produce better results.
>
> [7] Cheng, R., & Yan, J. (2021). On Joint Learning for Solving Placement and Routing in Chip Design. Advances in Neural Information Processing Systems, 34, 16508-16519.
>
> **Question 2 [comparisons on search time with baseline algorithms]**
>
> Answer: Thank you very much for your helpful advice. We provide a clock time comparison figure in Appendix A.7 Fig. 10 of our rebuttal version. You can see our method can achieve the best performance in a few hours.
>
> From the model's perspective, we only have CNN layers. There are a total of 14.59M parameters in our network, which is only slightly higher than the ResNet-18 model. As a result, our model is not complex and can achieve high efficiency.

---

> ### Author Response · Authors · 2022-08-02
> **Response to Reviewer #iBtL (Part 1/2)**
>
> Thank you very much for the constructive comments.
>
> About weaknesses:
>
> **Weakness 1 [simultaneous improvement of wirelength and density is counter-intuitive]**
>
> Response: Chip placement can be seen as an optimization problem with constraints like equation (1) in Section 2. Although we only optimize wirelength in the reward function, we can still meet the density constraint (no overlap constraint) with the aid of the position mask. Thus, we can guarantee no overlap in our placement results. According to the density definition in the "Density" paragraph of Appendix A.2, a placement without overlap has less density than placement with overlap. Thus we can improve density and wirelength simultaneously compared with methods that cannot meet the non-overlapping constraint.
>
> **Weakness 2 [lack of more details of how baselines are implemented and whether the comparisons are fair and experiments are trustable]**
>
> Response: For the baselines, including DREAMPlace, Graph Placement, and DeepPR, we implement them by their open repositories based on the default setting and our hyperparameter fine-tuning to optimize the results. We even got some better results compared with the original paper. For example, the wirelength can reach 31,426 for the benchmark *adaptec3*, as shown in Fig. 7, but it is 32,839 in the DeepPR paper [7]. Thus, our comparisons should be fair.
>
> Moreover, all experiment results are verified by the evaluation tool (plc_wrapper_main) provided by the Graph Placement repository [3], and the used benchmarks for different baselines and ours are the same.
>
> [7] Cheng, R., & Yan, J. (2021). On Joint Learning for Solving Placement and Routing in Chip Design. Advances in Neural Information Processing Systems, 34, 16508-16519.
>
> [3] https://storage.googleapis.com/rl-infra-public/circuit-training/placement_cost/plc_wrapper_main
>
> **Weakness 3 [few formulas]**
>
> Response: Thank you very much for your valuable advice. We have added the detailed description in the formula form of our method architecture in Appendix A.8 of our rebuttal version. You can also see them as follows.
>
> With current state $s_t$, we first calculate the position masks $f_p^t,f_p^{t+1}$, wire masks $f_w^t,f_w^{t+1}$ and view mask $f_v^t$ via the mask generation function $m(\cdot)$.
> $$f_p^t,f_p^{t+1},f_w^t,f_w^{t+1},f_v^t=m(s_t)$$
> Then we extract the local feature $z_t^l$ via local mask fusion $g_\omega(\cdot)$ and the global features $z_t^g$ via the global mask encoder $enc_\eta(\cdot)$.
> $$z_t^l=g_\omega(f_p^t,f_p^{t+1},f_w^t,f_w^{t+1})$$
> where $g_\omega(\cdot)$ is a 1×1 convolutional neural network with parameter $\omega$.
> $$z_t^g=enc_\eta(f_w^t,f_w^{t+1},f_v^t)$$
> where $enc_\eta(\cdot)$ is a convolutional neural network with ResNet-18 architecture with parameter $\eta$.
>
> With local features $z_t^l$ and global features $z_t^g$, the state value $\hat{V}_t$ is derived by
> $$\hat{V}_t=v_\phi(pos(t), z_t^g)$$
> where $v_\phi$ is an MLP-like neural network with parameter $\phi$ and $pos(t)$ is an embedding vector which is related to step $t$.
>
> We decode the global features $z_t^g$ into the dimension as same as the action space by the global mask decoder
> $$z_t^{'g}=dec_\delta(z_t^g)$$
> where $dec_\delta(\cdot)$ is a transposed convolutional neural network with parameter $\delta$.
>
> Finally, we concatenate the local features $z_t^l$ and global features $z_t^{'g}$ in the channel dimension and merge them by another 1×1 convolutional neural network $\psi_\xi(\cdot)$. We further combine it with the position mask $f_p^t$ to generate action $a_t$ via the policy network $\pi_\theta(\cdot)$
> $$a_t\sim\pi_\theta(\psi_\xi(z_t^l||z_t^{'g}),f_p^t)$$
> where $\pi_\theta(\cdot)$ is an MLP-like neural network with parameter $\theta$ and $\psi_\xi(\cdot)$ is a 1×1 convolutional neural network with parameter $\xi$.

---

> > ### Comment · Reviewer_iBtL · 2022-08-08
> > **Thanks for the response.**
> >
> > Thanks for the added formulation. The rebuttal addressed some of my questions. I have a followup questions:
> > 1) Have you compared the proposed ConvNet+MLP with a GNN [1] or a Transformer based network [2]? Is it the optimal combination? Intuitively, why it is better than GNN if the macro/standard cell placement map has a graphical structure?
> >
> > [1] "Chip Placement with Deep Reinforcement Learning", https://arxiv.org/abs/2004.10746
> > [2] "Transferable Graph Optimizers for ML Compilers", https://papers.nips.cc/paper/2020/file/9f29450d2eb58feb555078bdefe28aa5-Paper.pdf

---

> > > ### Author Response · Authors · 2022-08-08
> > > **Thanks for reply.**
> > >
> > > Thank you very much for your rapid reply.
> > >
> > > We have compared GNN-based methods, including Graph Placement (Edge-GNN based) [1] and DeepPR (GCN based) [2]. The corresponding comparison experiments can be seen in Table 2~7 in Section 4. Our method has advantages in different metrics and different circuits, which shows our method is the optimal combination. "Chip Placement with Deep Reinforcement Learning" [3] is just the arxiv version of the journal paper "A graph placement methodology for fast chip design" [1].
> > >
> > > Our method is better than GNN because the GNN-based model lost the pin offset information. The pin, not the node, directly affects the wirelength. And we gave an example in Appendix A.2 Figure 9, which shows the wirelength becomes 3x if we ignore the pin offset information. Also, it is challenging to learn the relationship between the graph structure and wirelength. Conversely, our proposed masks contain rich information for placement, including position, wire, and view, which help the model learn to find the optimal solution.
> > >
> > > For the transformer models, the number of their parameters is too large, which needs a large amount of data to train. It performs poorly in the online reinforcement learning tasks where the sample efficiency is significant, and we want to achieve better results within a limited data collection.
> > >
> > > [1] Mirhoseini, A., Goldie, A., Yazgan, M., Jiang, J. W., Songhori, E., Wang, S., ... & Dean, J. (2021). A graph placement methodology for fast chip design. Nature, 594(7862), 207-212.
> > >
> > > [2] Cheng, R., & Yan, J. (2021). On Joint Learning for Solving Placement and Routing in Chip Design. Advances in Neural Information Processing Systems, 34, 16508-16519.
> > >
> > > [3] Mirhoseini, A., Goldie, A., Yazgan, M., Jiang, J., Songhori, E., Wang, S., ... & Dean, J. (2020). Chip placement with deep reinforcement learning. arXiv preprint arXiv:2004.10746.

---

> > > > ### Comment · Reviewer_iBtL · 2022-08-09
> > > > **Thanks for your response.**
> > > >
> > > > In [1], node embedding and edge embeddings can be jointly learnt with a bi-directional GNN. I don't think anything you've mentioned, including the pin offset, cannot be captured by GNNs if the network is carefully designed.
> > > >
> > > > Thank you for updating the paper and addressing most of my questions. I will increase my score by one.
> > > >
> > > > [1] https://arxiv.org/abs/2004.10746

---

> > > > > ### Author Response · Authors · 2022-08-09
> > > > > **Thank you for your responsible reply**
> > > > >
> > > > > Thank you very much for updating and constructive suggestions. We will consider including the pin offset information in the graph in our future work.

---

> > > ### Author Response · Authors · 2022-08-09
> > > **Thanks for your efforts and look forward to your reply**
> > >
> > > We sincerely thank you for your efforts in reviewing our paper and your constructive suggestions again.
> > >
> > > We hope we have resolved all the concerns and showed the improved quality of the paper. And we deeply appreciate that if you could reconsider the score accordingly. We are always willing to address any of your further concerns.

---

> ### Author Response · Authors · 2022-08-08
> **Look forward to hearing from you**
>
> Dear reviewer #iBtL,
>
>
> Thank you very much for your precious review time and valuable comments. We have provided corresponding responses and results, which we wish could cover your concerns. We hope to further discuss with you whether or not your concerns have been addressed. Please let us know if you still have any unclear parts of our work. We are always ready to answer your concerns.

---

### Official Review · Reviewer_TA3J · 2022-07-10

**Rating:** 7
**Confidence:** 5
**Soundness:** 3 good
**Presentation:** 3 good
**Contribution:** 3 good

**Summary:**

The paper uses reinforcement learning to solve the chip placement problem. It views the placement canvas as a 2D image during the representation learning.


**Questions:**

1. Lines 192 - 201. The authors design the reward as the change of HPWL. How is the HPWL_t calculated if there are nodes not placed? For example, in a net with 3 pins on 3 nodes, what is the HPWL_t when we only place the first node? Since the other 2 nodes are not placed, it does not make sense to compute the HPWL for this net.
2. The authors use CNNs in representation learning. What about other neural architectures, such as Transformers?
3. What is the generation ability of the proposed method? Can it be used on unseen circuit benchmark?

**Limitations:**

The authors discuss the limitation at the end of the paper. The proposed method can only handle a small number of macros. It can not tackle a large number of movable instances, such as millions of standard cells. I think the authors should stress this limitation throughout the paper. Otherwise, readers may think that the paper proposes a method to place all instances. In the first three section, the authors use the term "placement" instead of "macro placement".

**Strengths And Weaknesses:**

Strengths
1. The paper is well written and organized. The appendix discusses related details.
2. The method is sound and well defined.
3. The results are charming and convincing.

Weaknesses
1. DREAMPlace is used as a baseline, which is not a good implementation for macro placement. The current DREAMPlace is majorly optimized for standard cell placement. It cannot handle macro placement very well. The authors may consider RePlAce [1] as the baseline.
[1] C. Cheng, A. B. Kahng, I. Kang and L. Wang, "RePlAce: Advancing Solution Quality and Routability Validation in Global Placement," in IEEE Transactions on Computer-Aided Design of Integrated Circuits and Systems, vol. 38, no. 9, pp. 1717-1730, Sept. 2019, doi: 10.1109/TCAD.2018.2859220.
2. The authors use ISPD 2005 benchmarks, which are for standard cell placement. Macros are fixed in these benchmarks. Authors should describe how they edit the benchmarks. The authors should also consider using MMS benchmarks [2].
[2] J. Z. Yan, N. Viswanathan and C. Chu, "Handling complexities in modern large-scale mixed-size placement," 2009 46th ACM/IEEE Design Automation Conference, 2009, pp. 436-441, doi: 10.1145/1629911.1630028.
3. Several benchmarks are missing, such as bigblue2, bigblue4. The authors should report the results of these benchmarks if they select the benchmark suites. Otherwise, readers may suspect that these results are not ideal.
4. The ablation study is not thorough. For instance, there are many masks in the proposed method. What is the contribution of each mask? Is every single mask necessary? What is the impact of the combination of these masks? Specifically, if the method is really great, it does not need the position mask that tells the agent the availability since the agent should have this information.

---

> ### Author Response · Authors · 2022-08-02
> **Response to Reviewer #TA3J**
>
> Thank you very much for your positive comments.
>
> About weaknesses:
>
> **Weakness 1 [add RePlAce as a baseline]**
>
> Response: Thank you very much for this helpful suggestion. We have updated the placement results of RePlAce method in Table 2 of our rebuttal version. You can also see it as follows.
>
> **Table 2. HPWL ($\times 10^5$) for method RePlAce**
>
> | Method  | adaptec1   | adaptec2     | adaptec3     | adaptec4   | bigblue1  | bigblue3     | ariane |
> |---------|------------|--------------|--------------|------------|-----------|--------------|--------|
> | RePlAce | 16.19±2.10 | 153.26±29.01 | 111.21±11.69 | 37.64±1.05 | 2.45±0.06 | 119.84±34.43 |  LG fail |
>
>
> **Weakness 2 [how to edit benchmark and the choice of benchmark]**
>
> Response: Thanks a lot for the constructive suggestion. Although macros are fixed in ISPD2005 benchmarks, we make all of them movable in our experiments like DeepPR paper which also uses this benchmark. The MMS benchmark is also to make most of the macros movable. Although they keep a few macros fixed according to the chip design, it is not directly related to methods. We will update the results of the MMS benchmark in our future work.
>
> **Weakness 3 [add more benchmarks in ISPD2005]**
>
> Response: Thanks a lot for the valuable suggestion. We have updated the results of the *bigblue4* benchmark in Appendix A.7 of our rebuttal version. And our MaskPlace also achieved the best performance in this circuit. You can also find the results as follows.
>
> **Table 3. HPWL ($\times 10^7$) results for *bigblue4* benchmark**
> | Benchmark | Random  | NTUPlace3 | RePlAce | DREAMPlace |
> |-----------|--------------|-----------------|--------------|------------|
> | bigblue4  | 128.06±3.94 | 48.38 | 11.80±0.73 | 12.29±1.64                          |
>
> | Benchmark | Graph Placement | DeepPR | DeepPR-no-overlap | MaskPlace  |
> |-----------|--------------|-----------------|--------------|------------|
> | bigblue4  | 53.35±4.06 | 68.30±4.44 | 115.08±2.29 | **11.07±0.90**  |
>
>
> **Weakness 4 [the ablation study is not thorough, add mask-related experiments]**
>
> Response: Thank you very much for the constructive suggestion. We have updated the ablation study figure by removing the wire mask and view mask in Fig. 6 of the rebuttal version.
> - Without the wire mask, we can find that the model can hardly converge. The reason is that it cannot estimate the wirelength when there is only position and view information. The reward cannot be optimized without wirelength information.
> - Without the view mask, the performance will decrease because the model loses the global placement information. However, it still has the wirelength information to keep a relatively short wirelength.
> - We do not remove the position mask because we use it to meet the no-overlap constraint, which is essential for a valid design. If we remove the position mask and add a related item to the reward, it will make the model difficult to train, and there is no guarantee that the result will meet the no-overlap constraint as well.
>
> About questions:
>
> **Question 1 [details about the calculation of $HPWL_t$]**
>
> Answer: We only consider the pins which have been placed when computing $HPWL_t$. For example, in a net with 3 pins on 3 modules, the pins are defined as $P_1$, $P_2$ and $P_3$. The $HPWL_t$ is the half parameter of the bounding box of $\{P_1\}$ if the pins $P_2$ and $P_3$ from the other two nodes have not been placed. The bounding box only contains one point; thus, the $HPWL_t$ equals 0 in this case.
>
> Similarly, when two nodes have been placed, the corresponding pins are $P_1$, $P_2$. The $HPWL_t$ is the half parameter of the bounding box of $\{P_1, P_2\}$. We disregard the $P_3$ because its position is uncertain.
>
>
> **Question 2 [what about other neural architectures]**
>
> Answer: For transformer models, the number of their parameters is too large, which needs more data to train. It performs poorly in the online reinforcement learning tasks where the sample efficiency is significant.
>
> For GCN-based models, it is still challenging to learn the relationship between the graph structure and wirelength, which has been proved in our experiments and explained in L62 of the introduction (the loss of pin offset information makes the inaccurate estimation of wirelength).
>
>
>
> **Question 3 [generation ability of the method for unseen circuits]**
>
> Answer: Thank you very much for your insightful question. Generally, the model trained on one benchmark can still work on an unseen benchmark with a little performance decrease. We have added one experiment for the model transferability in Appendix A.6 Table 16 of the rebuttal version for reference. You can also see it in Question 7 in the response for reviewer MdDg.
>
> For limitations:
>
> Thanks for your suggestion. We have added the "MaskPlace is mainly for macro placement due to the problem size." in L77 of Section 1 of our rebuttal version to stress our limitation.

---

### Official Review · Reviewer_f5N4 · 2022-07-10

**Rating:** 5
**Confidence:** 4
**Soundness:** 3 good
**Presentation:** 3 good
**Contribution:** 2 fair

**Summary:**

This paper presents a RL-based chip placement approach, MaskPlace, to automatically generate a valid chip layout design. Compared with the former approaches that apply hypergraph to represent the chip layout, MaskPlace adopts the pixel level graphical representation to represent the layout and pin offset, which allows for MaskPlace to produce a better performance. The results show that MaskPlace can achieve 60%-90% wirelength reduction with zero overlaps.

**Questions:**

I have the following questions for the paper:
1. Since MaskPlace uses the complicated 2D image as the input of the RL agents, I believe this will harm the convergence behavior of the MaskPlace compared with the other hypergraph-based approaches, as their input states will be much simpler. What's the convergence behavior of MaskPlace? What makes MaskPlace converge within a few hours?

2. How is the simulation is performed? Is it based on the real RTL synthesize netlist as [3]? If not, is it simulator based? Please describe the experiment settings in more detail.

3. MaskPlace eliminates the decision which violates the congestion constraint, why not include this congestion constraint in the reward function, so that a decision that violates the congestion constraint will produce a large negative reward?

**Limitations:**

NA.

**Strengths And Weaknesses:**

Strengths of the paper:
1. Chip placement with RL has been recently studied by multiple literature, this work proposes a better solution for chip placement. The comprehensive 2D pixelwise representation of the chip layout and wire length enables MaskPlace to involve the pin offset information when making the placement decision. This results in a better performance than the previous work which adopts hypergraph to represent the chip layout.

2. To generate the input state for the RL agents, heuristic algorithms are proposed to produce the input masks for the RL agents with relatively low complexity.

3. The presentation of the paper is clear, the experimental results are comprehensive and promising.

Weakness of the paper:

See the question section.

---

> ### Author Response · Authors · 2022-08-02
> **Response to Reviewer #f5N4**
>
> Thank you very much for the insightful comments.
>
> About questions:
>
> **Question 1 [why complicated 2D images do not harm the convergence]**
>
> Answer: Although the 2D image representation seems complicated, it provides more implied information related to the position and wirelength. For example, the wire mask can guide the agent on where to obtain shorter wirelength and help the agent find fewer good candidate positions to choose from. As a result, the convergence behavior of MaskPlace is from a random strategy to find better decisions based on masks accurately. Conversely, suppose we only use the hypergraph to describe the placement status. In that case, the model must learn the implied relationship between wirelength and the graph structure, which is a challenging task and will make the model difficult to converge. Also, the hypergraph representation will lose the pin offset information, as we discussed in Appendix A.2, further leading to a loss of accuracy in the wirelength estimation.
>
> **Question 2 [simulation setting and details of netlist]**
>
> Answer: We did not use commercial EDA tools for simulation, and our experiments are not commercial simulator based. Instead, we used the standard metrics in academia to evaluate placement, including wirelength, congestion, and density, as in Appendix A.2. They are also widely used in other works, such as NTUPlace3, DREAMPlace, Graph Placement, and DeepPR and are easier to reproduce. They can provide an accurate estimation of the performance of the final chip at a lower cost. Also, our results are verified by the evaluation tool (plc_wrapper_main) provided by the Graph Placement repository [3].
>
> Also, we use the public placement benchmarks, including ISPD2005 [4], IBM benchmark suite[5], and Ariane [6]. ISPD2005 and IBM benchmark suite benchmarks are directly derived from industrial ASIC designs, and the physical structure of these designs is completely preserved. Also, the Ariane design is derived from the real CVA6 RISC-V CPU. As a result, all our benchmarks are from real RTL synthesized netlists.
>
> [3] https://storage.googleapis.com/rl-infra-public/circuit-training/placement_cost/plc_wrapper_main
>
> [4] Nam, G. J., Alpert, C. J., Villarrubia, P., Winter, B., & Yildiz, M. (2005, April). The ISPD2005 placement contest and benchmark suite. In Proceedings of the 2005 international symposium on Physical design (pp. 216-220).
>
> [5] Adya, S. N., Chaturvedi, S., & Markov, I. L. (2009). ICCAD'04 mixed-size placement benchmarks. GSRC Bookshelf.
>
> [6] https://github.com/openhwgroup/cva6
>
> **Question 3 [why not include this congestion constraint in the reward function]**
>
> Answer: Compared with adding a large negative item to the reward, dealing with the congestion in the inference stage has the following advantages.
> - In the practical chip design scenario, one logic chip design needs to be adapted to different processes, which usually have different congestion thresholds $C_{th}$. In our method, we do not need to retrain the model. We only need to adjust the input of the congestion satisfaction module, thus effectively reducing design costs.
> - Adding one congestion item in the reward will introduce one weight hypermeter to adjust and make the reinforcement model difficult to optimize and converge. Also, it still cannot fully guarantee to meet the hard congestion constraint.
> - The model needs to calculate the congestion function for each training episode when adding it to the reward, affecting our training efficiency. Conversely, it just needs to compute congestion once in the inference stage by our method.

---

### Official Review · Reviewer_MdDg · 2022-07-17

**Rating:** 7
**Confidence:** 4
**Soundness:** 3 good
**Presentation:** 4 excellent
**Contribution:** 3 good

**Summary:**

Placement is one critical step to ensure circuit performance including power consumption, delay, and chip area in physical design. This paper presents a reinforcement-learning-based approach to optimize the half perimeter wirelength (HPWL), placing circuit modules sequentially. This proposed method adopts multi-view (mask) visual representation from the placement to ensure the non-overlapping property among modules. Each mask clearly captures a certain aspect (allowable positioning, metric change, global placement information) at the current placement snapshot, resulting in efficient policy learning. The paper performed an extensive experimental study on multiple benchmarks and demonstrated the effectiveness of individual components through the ablation study.

**Questions:**

1. For congestion satisfaction, the reviewer wonders what is the current reasoning behind this inconsistency in the training and inference phases, and if there is any way to mitigate this effect.
1. Detailed components clarification 1: How did the position embedding at time $t$ in the value network works? In addition, what is the purpose of this component?
1. Detailed components clarification 2: It seems that the ordering of the module for placement matters a lot according to Fig. 6. The reviewer wonders how the current sorting is done exactly. The authors said "*we follow previous works to sort the circuit modules ...*" in Training and Testing of Section 3. However, no corresponding literature is given. In addition, it would be appreciated if the authors can give more comments on the reasoning behind the sorting heuristics, and what is the consequence of bad ordering.
1. Detailed components clarification 3: The reviewer understands the power of looking more steps forward to the RL agent, and wonder how the step are generated here. In addition, it would be great if the authors can comment on the tradeoff between the computation efficiency and learning quality w.r.t. the number of steps looking ahead.
1. Detailed components clarification 4: It is unclear how the algorithm works if the module size doesn't fit the grid cell. For example, if a module occupies one grid cell without covering the entire grid, is it still possible to place another module in the same grid?
1. To the reviewer's understanding, the view mask contains only two values ${0,1}$. However, the view mask in Fig.3 contains three different colors. Is it a simple illustration error?
1. Table 7 gives the inference time for various methods. The reviewer is curious about how long the training takes. Moreover, do the training examples come from only a single placement task which is also used in inference, or multiple placement tasks which can be generalized to different tasks during the inference time?

**Limitations:**

The proposed methods can be hardly applied to standard cell placement due to the large state and action space. However, the authors show in Table 6 that the placement tool which can handle standard cell placement (like DREAMPlace) may also benefit from the proposed method of handling module placement. This is beneficial for the EDA tool development in general.
There is no identified negative societal impact.

**Strengths And Weaknesses:**

Strengths:
1. This paper transforms the geometric placement problem into multiple visual representations using three masks, which opens the possibility of using mature convolution networks to extract the global layout information. In addition, each mask has its distinct purpose, benefiting the overall placement task from different angles. The position mask guarantees the non-overlapping property of the proposed method; the wire mask gives a good estimation of the metric (HPWL) change for the action; and the view mask provides a global view of the current placement results.
1. The paper provides multiple components which can be beneficial for future research in this area. The authors designed multiple masks for the placement and their corresponding efficient generation algorithms, which can be used as individual components to inspire other related placement and routing tasks.
1. This paper conducts an extensive experimental study of the proposed methods over various benchmarks and multiple metrics against both state-of-the-art optimization-based and RL-based approaches, demonstrating the effectiveness of their method. In addition, the ablation study also shows the power of each individual component of the proposed method.

Weaknesses:
1. There exist certain inconsistencies between the training and inference phase due to congestion satisfaction. The congestion satisfaction is not considered during the RL training phase. However, it is used during the inference phase. During the inference phase, the probability matrix of the action doesn't anticipate the effect of the congestion satisfaction, which may result in lower quality placement compared to the training phase (ignoring the congestion constraint).
1. Some detailed components of the design RL frameworks are not fully explained, as listed in the questions below.

---

> ### Author Response · Authors · 2022-08-02
> **Response to Reviewer #MdDg (Part 2/2)**
>
> **Question 4 [reasons for the choice of the number of steps looking forward]**
>
> Answer: We have tried taking one, three, and more steps into consideration. If the model considers the next step only, the performance will decrease as "w/o $M^{t+1}$" in the ablation study in Fig. 6.
>
> Moreover, if we input the masks of three or more steps into our models, the performance will not increase further. Conversely, the efficiency will decrease, and the memory usage will increase. As a result, we choose to take the next two steps to keep the balance.
>
> **Question 5 [the situation when module size does not fit the grid cell]**
>
> Answer: As we mentioned in the "View Mask" paragraph in section 3 and "Compare to Graph Represent" paragraph in section 4, all our results use the ceiling function to compute the number of cells occupied by each module except the "MaskPlace (soft constraint)" in Table 3 using round function. If one grid cell has been partially or fully occupied, it will no longer be occupied by other modules. As a result, we can guarantee any two modules will not overlap on the chip.
>
> **Question 6 [inconsistency of the view mask value and Fig. 3]**
>
> Answer: Thanks a lot for your careful reading. It is an illustration error, and the view mask contains only two values, 0 and 1. We have fixed Fig. 3 in our rebuttal version.
>
> **Question 7 [details of training time & training examples]**
>
> Answer: Generally, the training time of our method is a few hours, and you can check Fig. 10 in the "Search time" paragraph in the Appendix A.7 of our rebuttal version for the exact search time.
>
> In our experiments, the training examples come from only a single placement benchmark which is also used in inference. But the trained model has a good transferability and can generalize to other benchmarks. You can check Table 16 in the "Transferability" paragraph in Appendix A.7 of our rebuttal version. You can also find the results as follows.
>
> **Table 1. HPWL ($\times 10^5$) results for transferability. The model has been trained on *adaptec1* benchmark and just took the inference in other benchmarks.**
>
> |       | adaptec2   | adaptec3   | adaptec4   | bigblue1  | bigblue3     | ariane     |
> |-------|------------|------------|------------|-----------|--------------|------------|
> | HPWL  | 85.56±9.41 | 89.77±6.72 | 87.32±3.93 | 2.87±0.31 | 160.63±10.41 | 19.32±2.02 |
> | ratio* | 1.16       | 1.06       | 1.11       | 1.20      | 1.76         | 1.32       |
>
> *Compared with the HPWL results from the model trained on the corresponding benchmark.

---

> ### Author Response · Authors · 2022-08-02
> **Response to Reviewer #MdDg (Part 1/2)**
>
> Thank you very much for your positive comments.
>
> About weaknesses:
>
> **Weakness 1 [inconsistency between the training and inference for congestion]**
>
> Response: The explanation can be seen in Question 1.
>
> **Weakness 2 [detailed components clarification]**
>
> Response: The clarification can be seen in Question 2 ~ Question 7 as follows.
>
> About questions:
>
> **Question 1 [inconsistency between the training and inference for congestion]**
>
> Answer: The wirelength will increase when people decrease the congestion threshold $C_{th}$ regardless of the method used because of the reduced feasible space. Compared with considering the congestion in the reward when training, our approach has many advantages:
>
> - In the practical chip design scenario, one logic chip design needs to be adapted to different processes, which usually have different congestion thresholds $C_{th}$. In our method, we do not need to retrain the model and adjust the input of the congestion satisfaction module, thus effectively reducing design costs.
> - Adding the congestion item to the reward will introduce one hyperparameter and make the reinforcement model challenging to optimize and converge. Also, it still cannot guarantee to meet the congestion constraint when this constraint is hard.
> - The model must calculate the congestion function for each training episode when adding it to the reward, decreasing efficiency. Conversely, it just needs to compute congestion once in the inference stage by our method.
>
> Although the probability matrix of actions cannot anticipate the effect of the congestion satisfaction in the inference, it implies a probability ranking for decision making. The congestion satisfaction module will remove impossible positions, and our method can still achieve the short wirelength in the remaining positions and satisfy the congestion threshold $C_{th}$ simultaneously.
>
> In the actual chip design scenario, we can abstract it into the optimization problems with constraints as equation (1) in Section 2. We do not need to continuously decrease the congestion item because it belongs to constraint items. So, it is reasonable to solve it in the inference stage.
>
>
> **Question 2 [how the position embedding at time t works, and what is the purpose]**
>
> Answer: The position embedding in MaskPlace is similar to the position embedding in the transformer. It converts the serial scalar $t$ into a 64 dimension vector. In the implementation, we use the torch.nn.Embedding from PyTorch, which is based on the lookup table and is learnable.
>
> The expected reward (wirelength) is different for different modules at different timesteps. Thus, the position embedding t helps the model build the connection between the step and the expected value more quickly.
>
> **Question 3 [how the ordering of modules is done, the related corresponding literature, the reasoning behind and what is the consequence of bad ordering]**
>
> Answer: Thanks a lot for reading carefully. "previous works" mainly refer to the Graph Placement method [2], and we have updated the citation in "Training and Testing" paragraph of Section 3 of the rebuttal version.
> Our method orders the modules by (1) area, (2) the number of nets, and (3) the number of connected modules that have been placed. We standardize these features and calculate a weighted score for each module. Then, we sort these modules based on their scores from largest to smallest.
>
> The reasons are as follows.
> - Placing modules with larger areas first can prevent placement failure due to not finding a sizable contiguous space to place big modules.
> - Modules connecting with more nets have more influence on the final wirelength. As a result, it should be considered to place first because there are more available positions for it.
> - For one module, if more modules connected to it have been placed on the chip, it is easier to place it because all these neighboring modules can provide wirelength and position information. Thus, the agent should make easy decisions first because we cannot change the position of placed modules in our method.
>
> If a good ordering is not applied, the performance will decrease, and the placement can even fail because some crucial modules cannot find suitable positions.
>
> [2] Mirhoseini, A., Goldie, A., Yazgan, M., Jiang, J. W., Songhori, E., Wang, S., ... & Dean, J. (2021). A graph placement methodology for fast chip design. Nature, 594(7862), 207-212.

---

### Author Response · Authors · 2022-08-02
**General Response: Contributions, New Experiments and Common Question**

We sincerely appreciate all reviewers' time and efforts in reviewing our paper. We are glad to find that reviewers generally recognized our contributions:

**Contributions.**

- **Model.** Introducing an innovative way to represent the chip by pixel-level masks to solve chip placement problems. [MdDg, f5N4, TA3J, iBtL] Multiple components are beneficial for future research in this area. [MdDg, f5N4, iBtL]

- **Experiment.** The experiments are comprehensive. [MdDg, f5N4, iBtL] Experiments showed that our method achieved better performance in all metrics without overlaps and that the designed chips are more realistic using our approach. [MdDg, f5N4, TA3J, iBtL]

- **Presentation.** The paper is well written and organized. [MdDg, f5N4, TA3J]

Also, we thank all reviewers for their valuable and constructive suggestions, which help us a lot in improving our paper. In addition to the pointwise responses below, we summarize supporting experiments added in our rebuttal version according to the reviewers' suggestions.

**New Experiments.**
- A new traditional optimization method baseline RePlAce [1] in Section 4, Table 2. [TA3J]
- The ablation experiments about different masks in Section 4, Fig. 6. [TA3J]
- The results from the new benchmark *bigblue4* in Appendix A.7, Table 15. [TA3J]
- The real search time comparison experiment in Appendix A.7, Fig. 10. [MdDg, iBtL]
- The model transferability experiment in Appendix A.7, Table 16. [MdDg, TA3J]


[1] Cheng, C. K., Kahng, A. B., Kang, I., & Wang, L. (2018). Replace: Advancing solution quality and routability validation in global placement. IEEE Transactions on Computer-Aided Design of Integrated Circuits and Systems, 38(9), 1717-1730.

**Common Question.**

Reviewers are interested in why we consider only one metric in the training reward but can still achieve the best performance in all three metrics and what motivates us to treat the metrics differently. [MdDg, f5N4, iBtL] Apart from the pointwise responses below, we summarize the answer here.

In the practical chip placement, we can abstract the problem into an optimization problem with constraints as equation (1) in Section 2. The wirelength is the optimization item, while the congestion and density are constraint items. Although we do not put these constraint items into the reward function, we deal with them by the position mask and congestion satisfaction module, making our results still have low congestion and density.

Compared with adding the congestion and density into the reward, our method has the following advantages.

- **For training.**
Adding more items to the reward will introduce more weight hyperparameters, and it is time-consuming to adjust them. Also, it will make the training more difficult to converge with a complex reward function. In addition, computing the congestion for each step also affects the training efficiency.

- **For congestion.**
Different chip processes have different congestion thresholds. Thus, we do not need to retrain the model if we want to apply one logic design to different chip processes. We need to adjust the threshold in the congestion satisfaction module only, which can help us save resources and design chips faster.

- **For congestion and density.**
In a typical chip design scenario, congestion and density belong to hard constraints, which means only when the constraints are met can the chip be guaranteed to be valid. If we use the penalty items in the reward, these constraints are still not guaranteed to be met. On the contrary, we can meet these hard constraints in our design.

---

### Meta-Review · Area_Chair_UyZG · 2022-08-26

**Recommendation:** Accept
**Confidence:** Certain

**Metareview:**

The reviewers are enthusiastic about the work, and all recommended for the acceptance of the paper. The reviewers think the work is solid and novel, and potentially impactful. For example, Reviewer MdDg noted "This paper transforms the geometric placement problem into multiple visual representations using three masks, which opens the possibility of using mature convolution networks to extract the global layout information." Thanks to the authors for the detail rebuttal and the thorough discussion with the reviewers. Incorporating these points raised in the communication will further improve the paper.

**Award:**

No

---

### Decision · Program_Chairs · 2022-09-14

Accept